# Learning Robust Statistics for Simulation-based Inference under Model Misspecification

**Daolang Huang**[*]
Aalto University
daolang.huang@aalto.fi

**Ayush Bharti**[*]
Aalto University
ayush.bharti@aalto.fi

**Amauri H. Souza**
Aalto University
Federal Institute of Ceará
amauri.souza@aalto.fi

**Luigi Acerbi**
University of Helsinki
luigi.acerbi@helsinki.fi

**Samuel Kaski**
Aalto University
University of Manchester
samuel.kaski@aalto.fi

## Abstract

Simulation-based inference (SBI) methods such as approximate Bayesian computation (ABC), synthetic likelihood, and neural posterior estimation (NPE) rely on simulating statistics to infer parameters of intractable likelihood models. However, such methods are known to yield untrustworthy and misleading inference outcomes under model misspecification, thus hindering their widespread applicability. In this work, we propose the first general approach to handle model misspecification that works across different classes of SBI methods. Leveraging the fact that the choice of statistics determines the degree of misspecification in SBI, we introduce a regularized loss function that penalizes those statistics that increase the mismatch between the data and the model. Taking NPE and ABC as use cases, we demonstrate the superior performance of our method on high-dimensional time-series models that are artificially misspecified. We also apply our method to real data from the field of radio propagation where the model is known to be misspecified. We show empirically that the method yields robust inference in misspecified scenarios, whilst still being accurate when the model is well-specified.

## 1 Introduction

Bayesian inference traditionally entails characterizing the posterior distribution of parameters assuming that the observed data came from the chosen model family [7]. However, in practice, the true data-generating process rarely lies within the family of distributions defined by the model, leading to *model misspecification*. This can be caused by, among other things, measurement errors or contamination in the observed data that are not included in the model, or when the model fails to capture the true nature of the physical phenomenon under study. Misspecified scenarios are especially likely for simulator-based models, where the goal is to describe some complex real-world phenomenon. Such simulators, also known as implicit generative models [25], have become prevalent in many domains of science and engineering such as genetics [71], ecology [3], astrophysics [40], economics [29], telecommunications [8], cognitive science [79], and agent-based modeling [87]. The growing field of *simulation-based inference* (SBI) [21, 51] tackles inference for such intractable likelihood models, where the approach relies on forward simulations from the model, instead of likelihood evaluations, to estimate the posterior distribution.

Traditional SBI methods include approximate Bayesian computation (ABC) [4, 51, 76], synthetic likelihood [67, 85] and minimum distance estimation [15]. More recently, the use of neural networks

---

[*]Equal contribution.

37th Conference on Neural Information Processing Systems (NeurIPS 2023).

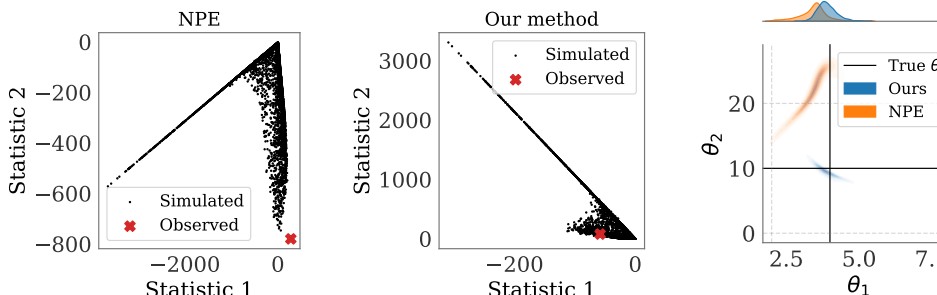

Figure 1: Inference on misspecified Ricker model with two parameters, see Section 4 for details. *Left:* Learned statistics obtained using NPE; each point (in black) corresponds to a parameter value sampled from the prior. The observed statistic (in red) is outside the set of statistics the model can simulate. *Middle:* Statistics learned using our robust method; the observed statistic is inside the distribution of simulated statistics. *Right:* The resulting posteriors obtained from NPE and our method. **The posterior obtained from our method is close to the true parameter value, while that from NPE is completely off, going outside the prior range** (denoted by dashed grey lines).

as conditional density estimators [72] has fuelled the development of new SBI methods that target either the likelihood (neural likelihood estimation) [52, 61, 75, 83], the likelihood-to-evidence ratio (neural ratio estimation) [22, 23, 28, 30, 39, 46, 77], or the posterior (neural posterior estimation) [38, 41, 53, 58, 59, 70, 75, 83]. A common feature of all these SBI methods is the choice of summary statistics of the data, which readily impacts their performance [14]. The summary statistics are either manually handcrafted by domain experts, or learned automatically from simulated data using neural networks [1, 17, 18, 26, 31, 47, 84].

The unreliability of Bayesian posteriors under misspecification is exacerbated for SBI methods as the inference relies on simulations from the misspecified model. Moreover, for SBI methods using conditional density estimators [5, 13, 41, 61, 69], the resulting posteriors can be wildly inaccurate and may even go outside the prior range [9, 16, 35]; see [27] for an instance of this problem in the study of molecules. This behavior is exemplified in Figure 1 for the posterior-targeting neural SBI method called *neural posterior estimation* (NPE). We see that when the model is misspecified, the NPE posterior goes outside the prior range. This is because the model is unable to match the observed statistic for any value of the parameters, as shown in Figure 1(left), and hence, NPE is forced to generalize outside its training distribution, yielding a posterior that is outside the prior range.

We argue that the choice of statistics is crucial to the notion of misspecification in SBI. Suppose we wish to fit a Gaussian model to data from some skewed distribution. If we choose, for instance, the sample mean and the skewness as statistics, then the Gaussian model would fail to match these statistics for any choice of parameters, and we would end up in the same situation as in Figure 1 where the SBI method is forced to generalize outside its training distribution. However, choosing the sample mean and sample variance as statistics instead may solve this issue, as the model may be able to replicate the chosen statistics for some value of parameters, even though the Gaussian model itself is misspecified for any skewed dataset. This problem of identifying a low-dimensional statistic of a high-dimensional dataset for which the model of interest is correctly specified is termed the *data selection problem* [82]. We refer to the statistics that solve the data selection problem as being *robust*.

**Contributions.** In this paper, we learn robust statistics using neural networks to solve the data selection problem. To that end, we introduce a regularized loss function that balances between learning statistics that are informative about the parameters, and penalizing those choices of statistics or features of the data that the model is unable to replicate. By doing so, the observed statistic does not go outside the set of statistics that the model can simulate, see Figure 1(middle), thereby yielding a posterior around the true parameter even under misspecification (Figure 1(right)). As our method relies on learning appropriate statistics and not on the subsequent inference procedure, it is applicable to all the statistics-based SBI methods, unlike other robust methods in the literature [24, 48, 81]. We substantiate our claim in Section 4 through extensive numerical studies on NPE and ABC — two fundamentally different SBI methods. Additionally in Section 5, we apply our method to a real case of model misspecification in the field of radio propagation [8].

**Related works.** Model misspecification has been studied in the context of different SBI methods such as ABC [9, 33, 35, 36, 74], synthetic likelihood [32, 34, 57], minimum distance estimation [24], and neural estimators [48, 81]. It was first analyzed for ABC methods such as the regression-adjustment ABC [5] in [35], and a robust approach was proposed in [33]. In [9], the authors proposed to handle model misspecification in ABC by incorporating the domain expert while selecting the statistics. Generalized Bayesian inference [12, 43, 49], which is a popular framework for designing methods that are robust to misspecification, has also been linked to ABC [74] and synthetic likelihood [57]. In [24], the authors propose a robust SBI method based on Bayesian nonparametric learning and the posterior bootstrap. More recently, robustness to model misspecification has been tackled for neural likelihood [48] and posterior estimators [81]. We use the latter method, called robust neural posterior estimation (RNPE) [81], as a baseline for comparing our proposed approach. Our method differs from these previous robust approaches in that it is applicable across distinct SBI methods.

## 2 Preliminaries

**Simulation-based inference (SBI).** Consider a simulator $\{\mathbb{P}_\theta : \theta \in \Theta\}$, which is a family of distributions parameterized by $\theta$ on data space $\mathcal{X} \subseteq \mathbb{R}^d$. We assume that $\mathbb{P}_\theta$ is intractable, but sampling independent and identically distributed (iid) data $\mathbf{x}_{1:n} = \{\mathbf{x}^{(1)}, \ldots, \mathbf{x}^{(n)}\} \sim \mathbb{P}_\theta$ is straightforward for $\mathbf{x}^{(i)} \in \mathcal{X}$, $1 \le i \le n$. Given iid observed data $\mathbf{y}_{1:n} \sim \mathbb{Q}$, $\mathbf{y}^{(i)} \in \mathcal{X}$, from some true data-generating process $\mathbb{Q}$ and a prior distribution of $\theta$, we are interested in estimating a $\theta^\star \in \Theta$ such that $\mathbb{P}_{\theta^\star} = \mathbb{Q}$, or alternatively, in characterizing the posterior distribution $p(\theta|\mathbf{y}_{1:n})$. Since $\mathcal{X}$ is a high-dimensional space in most practical cases, it is common practice to project both the simulated and the observed data onto a low-dimensional space of summary statistics $\mathcal{S}$ via a mapping $\eta : \mathcal{X}^n \to \mathcal{S}$, such that $\mathbf{s} = \eta(\mathbf{x}_{1:n})$ is the vector of simulated statistics, and $\mathbf{s}_{\text{obs}} = \eta(\mathbf{y}_{1:n})$ the observed statistic. In the following, we assume that this deterministic map $\eta$ is not known *a priori*, but needs to be learned using a neural network, henceforth called the *summary network* $\eta_\psi$, with trainable parameters $\psi$. We now introduce two different paradigms for learning the statistics.

**Learning statistics in neural posterior estimation (NPE) framework.** NPEs are a class of SBI methods which are becoming popular in astrophysics [40] and nuclear fusion [37, 80], among other fields. They map each statistic vector $\mathbf{s}$ to an estimate of the posterior $p(\theta|\mathbf{s})$ using conditional density estimators such as normalizing flows [41, 53, 59, 69]. The posterior is assumed to be a member of a family of distributions $q_\nu$, parameterized by $\nu$. NPEs map the statistics to distribution parameters $\nu$ via an *inference network* $h_\phi$, where $\phi$ constitutes the weights and biases of $h$, such that $q_{h_\phi(\mathbf{s})}(\theta) \approx p(\theta|\mathbf{s})$. The inference network is trained on dataset $\{(\theta_i, \mathbf{s}_i)\}_{i=1}^m$, simulated from the model using prior samples $\{\theta_i\}_{i=1}^m \sim p(\theta)$, by minimising the loss $\mathcal{L}(\phi) = -\mathbb{E}_{p(\theta, \mathbf{s})}[\log q_{h_\phi(\mathbf{s})}(\theta)]$. Once trained, the posterior estimate is then obtained by simply passing the observed statistic $\mathbf{s}_{\text{obs}}$ through the trained network. Hence, NPEs enjoy the benefit of amortization, wherein inference on new observed dataset is straightforward after a computationally expensive training phase. In cases where the statistics are not known *a priori*, the NPE framework allows for joint training of both the summary and the inference networks by minimizing the loss function

$$\mathcal{L}_{\text{NPE}}(\phi, \psi) = -\mathbb{E}_{p(\theta, \mathbf{x}_{1:n})} \left[ \log q_{h_\phi(\eta_\psi(\mathbf{x}_{1:n}))}(\theta) \right], \tag{1}$$

on the training dataset $\{(\theta_i, \mathbf{x}_{1:n,i})\}_{i=1}^m$ [69]. Even though NPEs are flexible and efficient SBI methods, they are known to perform poorly when the model is misspecified [16, 81]. This is because the observed statistic $\mathbf{s}_{\text{obs}}$ becomes an out-of-distribution sample under misspecification (see Figure 1 for an example), forcing the inference network in NPE to generalize outside its training distribution.

**Learning statistics using autoencoders.** Unlike NPEs, statistics for other SBI methods are learned prior to carrying out the inference procedure. This is achieved, for instance, by training an autoencoder with the reconstruction loss [1]

$$\mathcal{L}_{\text{AE}}(\psi, \psi_d) = \mathbb{E}_{p(\theta, \mathbf{x}_{1:n})} \left[ (\mathbf{x}_{1:n} - \tilde{\eta}_{\psi_d}(\eta_\psi(\mathbf{x}_{1:n})))^2 \right], \tag{2}$$

where $\psi$ and $\psi_d$ are the parameters of the encoder $\eta$ and the decoder $\tilde{\eta}$, respectively. The trained encoder $\eta_\psi$ is then taken as the summarizing function in the SBI method. In this paper, we will use the encoder $\eta_\psi$ to perform inference in an ABC framework, which we now recall.

**Approximate Bayesian computation (ABC).** ABC is arguably the most popular SBI method, and is widely used in many research domains [64]. ABC relies on computing the distance between the simulated and the observed statistics to obtain samples from the approximate posterior of a simulator. Given a discrepancy $\rho(\cdot, \cdot)$ and a tolerance threshold $\delta$, the basic rejection-ABC method involves repeating the following algorithm: (i) sample $\theta' \sim p(\theta)$, (ii) simulate $\mathbf{x}'_{1:n} \sim \mathbb{P}_{\theta'}$, (iii) compute $\mathbf{s}' = \eta(\mathbf{x}'_{1:n})$, and (iv) if $\rho(\mathbf{s}', \mathbf{s}_{\mathrm{obs}}) < \delta$, accept $\theta'$, until a sample $\{\theta'_i\}_{i=1}^{n_\delta}$ is obtained from the approximate posterior. Conditional density estimators have also been used in ABC to improve the posterior approximation of the rejection-ABC method [5, 11, 13, 14]. These so-called regression adjustment ABC methods involve fitting a function $g$ between the accepted parameters and statistics pairs $\{(\theta'_i, \mathbf{s}'_i)\}_{i=1}^{n_\delta}$ as $\theta'_i = g(\mathbf{s}'_i) + \omega_i, 1 \le i \le n_\delta$, where $\{\omega_i\}_{i=1}^{n_\delta}$ are the residuals. Once fitted, the accepted parameter samples are then adjusted as $\tilde{\theta}'_i = \hat{g}(\mathbf{s}_{\mathrm{obs}}) + \hat{\omega}_i$, where $\hat{g}(\mathbf{s})$ is the estimate of $\mathbb{E}[\theta|\mathbf{s}]$ and $\hat{\omega}_i$ is the empirical residual. Despite the popularity of the regression adjustment ABC methods, they have also been shown to behave wildly under misspecification [9, 35], similar to NPEs.

## 3   Methodology

**Model misspecification in SBI.** Bayesian inference assumes that there exists a $\theta^\star \in \Theta$ such that $\mathbb{P}_{\theta^\star} = \mathbb{Q}$. Under model misspecification, $\mathbb{P}_\theta \ne \mathbb{Q}$ for any $\theta \in \Theta$, which can lead to unreliable predictive distributions. However, this definition does not apply for models whose inference is carried out in light of certain summary statistics. For instance, a Gaussian model is clearly misspecified for a bimodal data sample. However, if inference is based on, say, the sample mean and the sample variance of the data, the model may still match the observed statistics accurately. Hence, even when a simulator is misspecified with respect to the true data-generating process, it may still be well-specified with respect to the observed statistic. The choice of statistics therefore dictates the notion of misspecification in SBI. We propose an analogous definition for misspecification in SBI:

**Definition 3.1** (**Misspecification of model-summarizer pair in SBI**). *Let $\eta_\# \mathbb{P}_\theta^n$ be a pushforward of the probability measure $\mathbb{P}_\theta^n = \mathbb{P}_\theta \times \cdots \times \mathbb{P}_\theta$ on $\mathcal{X}^n$ under $\eta : \mathcal{X}^n \to \mathcal{S}$, meaning that for $A \subset \mathcal{S}$, $\eta_\# \mathbb{P}_\theta^n(A) = \mathbb{P}_\theta^n(\eta^{-1}(A)) = \mathbb{P}_\theta(\eta^{-1}(A)_1) \times \cdots \times \mathbb{P}_\theta(\eta^{-1}(A)_n)$. Then, $\{\eta_\# \mathbb{P}_\theta^n : \theta \in \Theta\}$ is a set of distributions on the statistics space $\mathcal{S}$ induced by the model for a given summarizer $\eta$, and $\eta_\# \mathbb{Q}^n$ is the distribution of the statistics from the true data-generating process. Then, the model-summarizer pair is misspecified if, $\forall \theta \in \Theta$, $\eta_\# \mathbb{Q}^n \ne \eta_\# \mathbb{P}_\theta^n$.*

It is trivial to see that if the model is well-specified in the data space $\mathcal{X}$, it will also be well-specified in the statistic space $\mathcal{S}$ for any choice of $\eta$. The converse, however, is not true, as previously mentioned. Using this definition, we quantify the level of misspecification of a model-summarizer pair as follows:

**Definition 3.2** (**Misspecification margin in SBI**). *Let $\mathcal{D}$ be a distance defined on the set of all Borel probability measures on $\mathcal{S}$. Then, for a given $\eta$, we define the misspecification margin $\varepsilon_\eta$ in SBI as*

$$\varepsilon_\eta = \inf_{\theta \in \Theta} \mathcal{D}\left(\eta_\# \mathbb{P}_\theta^n, \eta_\# \mathbb{Q}^n\right). \tag{3}$$

Note that the margin is equal to zero for the well-specified case. Moreover, the larger the margin, the bigger the mismatch between the simulated and the observed statistics w.r.t $\eta$. Our aim is to learn an $\eta$ such that $\mathbb{P}_\theta$ is represented well whilst the model-summarizer pair is no longer misspecified as per Definition 3.1, or alternatively, has zero margin as per Definition 3.2. We formulate this as a constrained optimization problem in Equation (6).

**Learning robust statistics for SBI.** Suppose we have the flexibility to choose the summarizer $\eta$ from a family of functions parameterized by $\psi$. Then, a potential approach for tackling model misspecification in SBI is to select the summarizer $\eta$ that minimizes the misspecification margin $\varepsilon_\eta$. However, that involves computing the infimum for all possible choices of $\psi$. To avoid that, we minimize an upper bound on the margin instead, given as

$$\varepsilon_{\eta_\psi} \le \varepsilon_{\eta_\psi}^{\mathrm{upper}} = \mathbb{E}_{p(\theta)}\left[\mathcal{D}\left(\eta_{\psi\#} \mathbb{P}_\theta^n, \eta_{\psi\#} \mathbb{Q}^n\right)\right]. \tag{4}$$

Note that Equation (4) is valid for any choice of $p(\theta)$, and we call $\varepsilon_{\eta_\psi}^{\mathrm{upper}}$ the *margin upper bound*.

Minimizing the margin upper bound alone would lead to $\eta$ being a constant function, which when used for inference, would yield the prior distribution. Hence, there is a trade-off between choosing an $\eta$ that is informative about the parameters and an $\eta$ that minimizes the margin upper bound. We

propose to navigate this trade-off by including the margin upper bound as a regularization term in the loss function for learning $\eta$. To that end, let $\mathcal{L}(\omega, \psi)$ be the loss function used to learn $\eta_\psi$ (along with additional parameters $\omega$) which, for instance, can be either $\mathcal{L}_{\text{NPE}}$ from Equation (1) or $\mathcal{L}_{\text{AE}}$ from Equation (2). Then, our proposed loss with robust statistics (RS) is written as

$$\mathcal{L}_{\text{RS}}(\omega, \psi) = \mathcal{L}(\omega, \psi) + \lambda \underbrace{\mathbb{E}_{p(\theta)}\left[\mathcal{D}\left(\eta_{\psi\#}\mathbb{P}_\theta^n, \eta_{\psi\#}\mathbb{Q}^n\right)\right]}_{\text{regularization}}, \tag{5}$$

where $\lambda \geq 0$ is the weight we place on the regularization term relative to the standard loss $\mathcal{L}$. Minimizing the loss function in Equation (5) corresponds to solving the Lagrangian relaxation of the optimization problem:

$$\min_{\omega, \psi} \quad \mathcal{L}(\omega, \psi) \tag{6}$$
$$\text{s.t.} \quad \mathbb{E}_{p(\theta)}\left[\mathcal{D}\left(\eta_{\psi\#}\mathbb{P}_\theta^n, \eta_{\psi\#}\mathbb{Q}^n\right)\right] \leq \xi, \quad \xi > 0,$$

with $\lambda \geq 0$ being the Lagrangian multiplier fixed to a constant value.

**Estimating the margin upper bound.** In order to implement the proposed loss from Equation (5), we need to estimate the margin upper bound using the training dataset $\{(\theta_i, \mathbf{x}_{1:n,i})\}_{i=1}^m \sim p(\theta, \mathbf{x}_{1:n})$ sampled from the prior and the model. However, we only have one sample each from the distributions $\eta_{\psi\#}\mathbb{P}_{\theta_i}^n, i = 1, \ldots, m$, and one sample of the observed statistic from $\eta_{\psi\#}\mathbb{Q}^n$. Taking $\mathcal{D}$ to be the Euclidean distance $\|\cdot\|$, we can estimate the margin upper bound as $\frac{1}{m}\sum_{i=1}^m \|\eta_\psi(\mathbf{x}_{1:n,i}) - \eta_\psi(\mathbf{y}_{1:n})\|$. Although easy to compute, we found this choice of $\mathcal{D}$ to yield overly conservative posteriors even in the well-specified case, see Appendix B.3 for the results. This is because the Euclidean distance is large even if only a handful of simulated statistics — corresponding to different parameter values in the training data — are far from the observed statistic. As a result, the regularizer dominates the loss function while the standard loss term $\mathcal{L}$ related to $\theta$ is not minimized, leading to underconfident posteriors. Hence, we need a distance $\mathcal{D}$ that is robust to outliers and can be computed between the set $\{\eta_\psi(\mathbf{x}_{1:n,i})\}_{i=1}^m$ and $\eta_\psi(\mathbf{y}_{1:n})$, instead of computing the distance point-wise.

To that end, we take $\mathcal{D}$ to be the maximum mean discrepancy (MMD) [42], which is a notion of distance between probability distributions or datasets. MMD has a number of attractive properties that make it suitable for our method: (i) it can be estimated efficiently using samples [15], (ii) it is robust against a few outliers (unlike the KL divergence), and (iii) it can be computed for unequal sizes of datasets. The MMD has been used in a number of frameworks such as ABC [8, 10, 50, 54, 62], minimum distance estimation [2, 15, 19, 56], generalized Bayesian inference [20, 57], and Bayesian nonparametric learning [24]. Similar to our method, the MMD has previously been used as a regularization term to train Bayesian neural networks [66], and in the NPE framework to detect model misspecification [73]. While we include the mmD-based regularizer to ensure that the observed statistic is not an out-of-distribution sample in the summary space, the regularizer in [73] involves computing the MMD between the simulated statistics and samples from a standard Gaussian, thus ensuring that the learned statistics are jointly Gaussian. They then conduct a goodness-of-fit test [42] to detect if the model is misspecified. Their method is complementary to ours, such that our method can be used once misspecification has been detected using [73].

**Maximum mean discrepancy (MMD) as $\mathcal{D}$.** The MMD is a kernel-based distance between probability distributions, computed by mapping the distributions to a function space called the reproducing kernel Hilbert space (RKHS). Let $\mathcal{H}_k$ be the RKHS associated with the symmetric and positive definite function $k : \mathcal{S} \times \mathcal{S} \to \mathbb{R}$, called a reproducing kernel [6], defined on the space of statistics $\mathcal{S}$ such that $f(\mathbf{s}) = \langle f, k(\cdot, \mathbf{s})\rangle_{\mathcal{H}_k} \forall f \in \mathcal{H}_k$ (called the reproducing property). We denote the norm and inner product of $\mathcal{H}_k$ by $\|\cdot\|_{\mathcal{H}_k}$ and $\langle\cdot, \cdot\rangle_{\mathcal{H}_k}$, respectively. Any distribution $\mathbb{P}$ can be mapped to $\mathcal{H}_k$ via its kernel-mean embedding $\mu_{k,\mathbb{P}} = \int_{\mathcal{S}} k(\cdot, \mathbf{s})\mathbb{P}(d\mathbf{s})$. The MMD between two arbitrary probability measures $\mathbb{P}$ and $\mathbb{Q}$ on $\mathcal{S}$ is defined as the distance between their embeddings in $\mathcal{H}_k$, i.e., $\text{MMD}_k(\mathbb{P}, \mathbb{Q}) = \|\mu_{k,\mathbb{P}} - \mu_{k,\mathbb{Q}}\|_{\mathcal{H}_k}$ [55]. Using the reproducing property, we can express the squared-MMD as

$$\text{MMD}_k^2[\mathbb{P}, \mathbb{Q}] = \mathbb{E}_{\mathbf{s}, \mathbf{s}' \sim \mathbb{P}}[k(\mathbf{s}, \mathbf{s}')] - 2\mathbb{E}_{\mathbf{s} \sim \mathbb{P}, \mathbf{s}' \sim \mathbb{Q}}[k(\mathbf{s}, \mathbf{s}')] + \mathbb{E}_{\mathbf{s}, \mathbf{s}' \sim \mathbb{Q}}[k(\mathbf{s}, \mathbf{s}')], \tag{7}$$

which is computationally convenient as the expectations of the kernel can be estimated using iid samples from $\mathbb{P}$ and $\mathbb{Q}$. Given samples of simulated statistics $\{\eta_\psi(\mathbf{x}_{1:n,i})\}_{i=1}^l$ and the observed

statistic $\eta_\psi(\mathbf{y}_{1:n})$, we estimate the squared-MMD between them using the V-statistic estimator [42]:

$$\text{MMD}_k^2[\eta_{\psi\#}\mathbb{P}_\theta^n, \eta_{\psi\#}\mathbb{Q}^n] \approx \text{MMD}_k^2[\{\eta_\psi(\mathbf{x}_{1:n,i})\}_{i=1}^l, \eta_\psi(\mathbf{y}_{1:n})] \tag{8}$$

$$= \frac{1}{l^2} \sum_{i,j=1}^l k(\eta_\psi(\mathbf{x}_{1:n,i}), \eta_\psi(\mathbf{x}_{1:n,j})) - \frac{2}{l} \sum_{i=1}^l k(\eta_\psi(\mathbf{x}_{1:n,i}), \eta_\psi(\mathbf{y}_{1:n})).$$

Note that the last term in Equation (7) is always constant for the estimator above as we only have one data-point of the observed statistic, hence we disregard it. Equation (8) therefore corresponds to estimating the MMD between the distribution of the simulated statistics for a given $\psi$ and a Dirac measure on $\eta_\psi(\mathbf{y}_{1:n})$. The computational cost of estimating the squared-MMD is $\mathcal{O}(l^2)$ and its rate of convergence is $\mathcal{O}(l^{-\frac{1}{2}})$. The NPE loss with robust statistics (NPE-RS) can then be estimated as

$$\hat{\mathcal{L}}_{\text{NPE-RS}}(\phi, \psi) = -\frac{1}{m} \sum_{i=1}^m \log q_{h_\phi(\eta_\psi(\mathbf{x}_{1:n,i}))}(\theta_i) + \lambda \text{MMD}_k^2\left[\{\eta_\psi(\mathbf{x}_{1:n,i})\}_{i=1}^l, \eta_\psi(\mathbf{y}_{1:n})\right]. \tag{9}$$

Similarly, the autoencoder loss with robust statistics (AE-RS) reads

$$\hat{\mathcal{L}}_{\text{AE-RS}}(\psi, \psi_d) = \frac{1}{m} \sum_{i=1}^m (\mathbf{x}_{1:n,i} - \tilde{\eta}_{\psi_d}(\eta_\psi(\mathbf{x}_{1:n,i})))^2 + \lambda \text{MMD}_k^2\left[\{\eta_\psi(\mathbf{x}_{1:n,i})\}_{i=1}^l, \eta_\psi(\mathbf{y}_{1:n})\right].$$
$$\tag{10}$$

We use a subset of the training dataset of size $l < m$ to compute the MMD instead of $m$ to avoid incurring additional computational cost in case $m$ is large. The MMD-based regularizer can also be used as a score in a classification task to detect model misspecification, as shown in Appendix A.

**Role of the regularizer $\lambda$.** The regularizer $\lambda$ penalizes learning those statistics for which the observed statistic $\eta_\psi(\mathbf{y}_{1:n})$ is far from the set of statistics that the model can simulate given a prior distribution. When $\lambda$ tends to zero, maximization of the likelihood dictates learning of both $\phi$ and $\psi$ in Equation (9), and our method converges to the NPE method. On the other hand, the regularization term is minimized when the summary network outputs the same statistics for both simulated and observed data, i.e., when $\mathcal{D}$ is zero. In this case, the inference network can only rely on information from the prior $p(\theta)$. As a result, for large values of $\lambda$, we expect the regularization term to dominate the loss and the resulting posterior to converge to the prior distribution. Similar argument holds for the autoencoder loss in Equation (10). We empirically observe this behavior in Section 4. Hence, $\lambda$ encodes the trade-off between efficiency and robustness of our inference method. Choosing $\lambda$ can be cast as a hyperparameter selection problem, for which we can leverage additional data (if available) as a validation dataset, or use post-hoc qualitative analysis of the posterior predictive distribution.

## 4    Numerical experiments

We apply our method of learning robust statistics to two different SBI frameworks — NPE [41] and ABC [5] (see Appendix B.6 for results on applying our method to neural likelihood estimator [52]). We also compare the performance of our method against RNPE [81], by using the output of the trained summary network in NPE as statistics for RNPE. Experiments are conducted on synthetic data from two time-series models, namely the Ricker model from population ecology [67, 85] and the Ornstein-Uhlenbeck process [18]. Real data experiment on a radio propagation model is presented in Section 5. Analysis of the computational cost of our method and results on a 10-dimensional Gaussian model is presented in Appendix B.4 and B.5, respectively. The code to reproduce our experiments is available at https://github.com/huangdaolang/robust-sbi.

**Ricker model**    simulates the evolution of population size $N_t$ over the course of time $t$ as $N_{t+1} = \exp(\theta_1)N_t \exp(-N_t + e_t)$, $t = 1, \ldots, T$, where $\exp(\theta_1)$ is the growth rate parameter, $e_t$ are zero-mean iid Gaussian noise terms with variance $\sigma_e^2$, and $N_0 = 1$. The observations $x_t$ are assumed to be Poisson random variables such that $x_t \sim \text{Poiss}(\theta_2 N_t)$. For simplicity, we fix $\sigma_e^2 = 0.09$ and estimate $\theta = [\theta_1, \theta_2]^\top$ using the prior distribution $\mathcal{U}([2, 8] \times [0, 20])$, and $T = 100$ time-steps.

**Ornstein-Uhlenbeck process (OUP)**    is a stochastic differential equation model widely used in financial mathematics and evolutionary biology. The OU process $x_t$ is defined as:

$$x_{t+1} = x_t + \Delta x_t, \quad t = 1, \ldots, T,$$
$$\Delta x_t = \theta_1[\exp(\theta_2) - x_t]\Delta t + 0.5w,$$

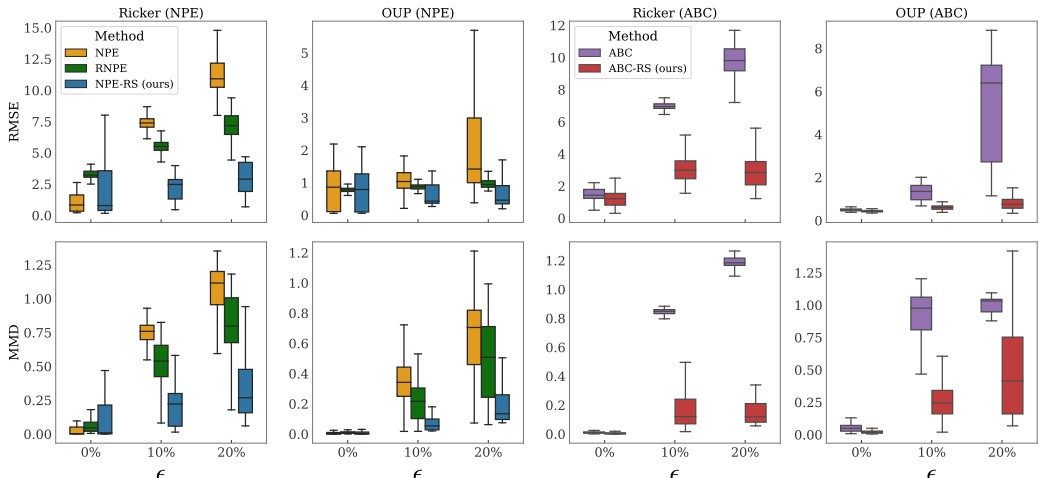

Figure 2: Performance of the SBI methods in terms of RMSE and MMD for both the Ricker model and OUP. Each box represents the median and interquartile range (IQR), while the whiskers extend to the furthest points within 1.5 times the IQR from the edges of the box. For the well-specified case ($\epsilon = 0\%$), the proposed NPE-RS and ABC-RS methods perform similar to their counterpart NPE and ABC, respectively. Under misspecification ($\epsilon > 0\%$), NPE-RS and ABC-RS achieve lower RMSE and MMD values, demonstrating robustness to model misspecification.

where $T = 25$, $\Delta t = 0.2$, $x_0 = 10$, and $w \sim \mathcal{N}(0, \Delta t)$. A uniform prior $\mathcal{U}([0, 2] \times [-2, 2])$ is placed on the parameters $\theta = [\theta_1, \theta_2]^\top$.

**Contamination model.** We test for robustness relative to the $\epsilon$-contamination model [65], similar to [24]. For both the Ricker model and the OUP, the observed data comes from the distribution $\mathbb{Q} = (1 - \epsilon)\mathbb{P}_{\theta_{\text{true}}} + \epsilon\mathbb{P}_{\theta_c}$, where a large proportion of the data comes from the model with $\theta_{\text{true}}$, while $\epsilon$ proportion of the data comes from the distribution $\mathbb{P}_{\theta_c}$. Hence, $\epsilon \in [0, 1]$ denotes the level of misspecification within both the models, with $\epsilon = 0$ resulting in the well-specified case. We take $\theta_{\text{true}} = [0.5, 1.0]^\top$, $\theta_c = [-0.5, 1.0]^\top$ for OUP, and $\theta_{\text{true}} = [4, 10]^\top$, $\theta_c = [4, 100]^\top$ for the Ricker model.

**Implementation details.** We take $m = 1000$ samples for the training data and $n = 100$ realizations of both the observed and simulated data for each $\theta$. We set $\lambda$ using an additional dataset simulated from the models using $\theta_{\text{true}}$. For the MMD, we take the kernel to be the exponentiated-quadratic kernel $k(\mathbf{s}, \mathbf{s}') = \exp(-\|\mathbf{s} - \mathbf{s}'\|_2^2/\beta^2)$, and set its lengthscale $\beta$ using the median heuristic $\beta = \sqrt{\text{med}/2}$ [42], where med denotes the median of the set of squared two-norm distances $\|\mathbf{s}_i - \mathbf{s}_j\|_2^2$ for all pairs of distinct data points in $\{\mathbf{s}_i\}_{i=1}^m$. We use $l = 200$ samples of the simulated statistics to estimate the MMD. For the Ricker model, the summary network $\eta_\psi$ is composed of 1D convolutional layers, whereas for the OUP, $\eta_\psi$ is a combination of bidirectional long short-term memory (LSTM) recurrent modules and 1D convolutional layers. The dimension of the statistic space is set to four for both the models. We take $q$ to be a conditional normalizing flow for all the three NPE methods. We take the tolerance $\delta$ in ABC to be the top 5% of samples that yield the smallest distance.

**Performance metrics.** We evaluate the accuracy of the posterior, as well as the posterior predictive distribution of the SBI methods. For the posterior distribution, we compute the root mean squared error (RMSE) as $(1/N \sum_{i=1}^N (\theta_i - \theta_{\text{true}})^2)^{\frac{1}{2}}$ where $\{\theta_i\}_{i=1}^N$ are posterior samples. For the predictive accuracy, we compute the MMD between the observed data and samples from the posterior predictive distribution of each method. As the models simulate high-dimensional data, we use the kernel specialized for time-series from [8]. The lengthscale of this kernel is set to $\beta = 1$ for all misspecification levels to facilitate fair comparison.

**Results.** Figure 2 presents the results for both the Ricker model and the OUP across 100 runs. We observe that both NPE and ABC with our robust statistics (RS) outperform their counterparts, including RNPE, under misspecification ($\epsilon = 10\%$ and $20\%$). Moreover, our performance is similar

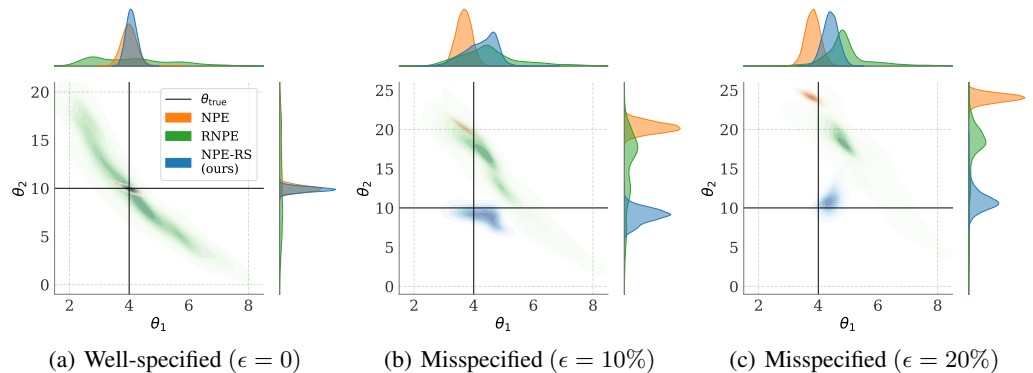

(a) Well-specified ($\epsilon = 0$)  (b) Misspecified ($\epsilon = 10\%$)  (c) Misspecified ($\epsilon = 20\%$)

Figure 3: Posteriors obtained from NPE, RNPE, and our NPE-RS method for the Ricker model. We perform similar to NPE in the well-specified case, unlike RNPE. Under misspecification, NPE and RNPE posteriors drift away from $\theta_{\text{true}}$, going even beyond the prior range (denoted by dashed gray lines) in the case of NPE. **Our method is robust to model misspecification**.

to NPE in the well-specified case ($\epsilon = 0\%$). An instance of the NPE posteriors is shown in Figure 3 for the Ricker model. Our NPE-RS posterior is very similar to the NPE posterior for the well-specified case, whereas the RNPE posterior is underconfident, as noted in [81]. Although RNPE is more robust than NPE under misspecification, its posterior still drifts away from $\theta_{\text{true}}$, while NPE-RS posterior stays around $\theta_{\text{true}}$ even for $\epsilon = 20\%$. This is because our method has the flexibility to choose appropriate statistics based on misspecification level, while RNPE is bound to a fixed choice of pre-selected statistics. Posterior plots for ABC and OUP are given in Appendix B.2.

**Prior misspecification.** So far our discussion has pertained to the case of likelihood misspecification. However, in Bayesian inference, the prior is also a part of the model. Hence, a potential form of misspecification is *prior misspecification*, in which the prior gives low or even zero probability to the "true" data generating parameters. We investigate the issue of prior misspecification using the Ricker model as an example. To that end, we modify the prior distribution of $\theta_2$ by setting it to $\text{LogNormal}(0.5, 1.0)$ while keeping the prior of $\theta_1$ unchanged. We use $\theta_{\text{true}} = [4, 25]^\top$ to generate the observed data, as $\theta_2 = 25$ has very low density under the lognormal prior, and the probability that a value of $\theta_2 \geq 25$ will be sampled is 0.00372. The result in Figure 4 shows that NPE and RNPE yield posteriors that do not include the true value of $\theta_2$, whereas our NPE-RS posterior is still around $\theta_{\text{true}}$. This highlights the effectiveness of our method to handle cases of prior misspecification as well.

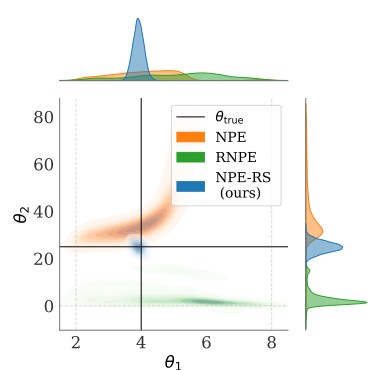

Figure 4: Posteriors from NPE, RNPE, and NPE-RS for Ricker model. **NPE-RS is robust to prior misspecification**.

**Varying regularizer $\lambda$.** As mentioned in Section 3, our method converges to NPE as $\lambda$ tends to zero, and to the prior distribution as $\lambda$ becomes high. To investigate this effect, we vary $\lambda$ and measure the distance between the NPE-RS and the NPE posteriors, and between the NPE-RS posterior and the prior. We use the MMD from Equation (7) to measure this distance on the Ricker model in the misspecified case ($\epsilon = 20\%$). We see in Figure 5(left) that the MMD values between the prior and the NPE-RS posteriors go close to zero for $\lambda = 10^3$, indicating that our method essentially returns the prior for high values of $\lambda$. On the other hand, the MMD values between NPE and NPE-RS posteriors in Figure 5(middle) are smallest for $\lambda = 0.01$ and $\lambda = 0.1$. For non-extreme values of $\lambda$ (i.e. $\lambda = 1$ or 10), we observe maximum difference between NPE-RS and NPE. This is because the NPE posteriors tend to move out of the prior support under misspecification, whereas the NPE-RS posteriors remain around $\theta_{\text{true}}$, as shown in Figure 3. Finally, the NPE-RS posteriors are also close to the NPE posteriors for small values of $\lambda$ in the well-specified case, as shown in Figure 5(right).

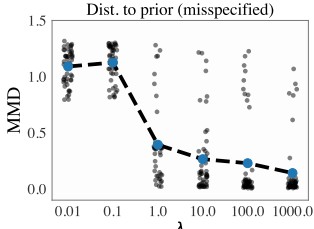 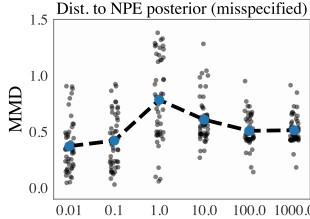 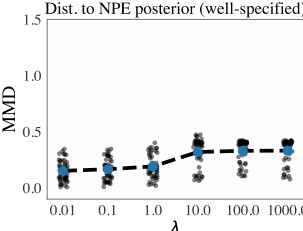

Figure 5: MMD between the NPE-RS posteriors and *(left)* the prior in the misspecified setting, *(middle)* the NPE posteriors in the misspecified setting, *(right)* the NPE posteriors in the well-specified setting, for different values of $\lambda$.

## 5 Application to real data: A radio propagation example

Our final experiment involves a stochastic radio channel model, namely the Turin model [45, 63, 78], which is used to simulate radio propagation phenomena in order to test and design wireless communication systems. The model is driven by an underlying point process which is unobservable, thereby rendering its likelihood function intractable. The model simulates high-dimensional complex-valued time-series data, and has four parameters. The parameters govern the starting point, the slope of the decay, the rate of the point-process, and the noise floor of the time-series, as shown in Figure 6.

We attempt to fit this model to a real dataset from a measurement campaign [44] for which the model is known to be misspecified [8], on account of the data samples being non-iid. The averaged power of the data (square of the absolute value) as a function of time is shown by the black curve in Figure 6. The data dimension is 801 and we have $n = 100$ realizations. We take the power of the complex data in decibels[*] as input to the summary network, which consists of 1D convolutional layers, and we set uniform priors for all the four parameters. Descriptions of the model, the data, and the prior ranges are provided in Appendix C.

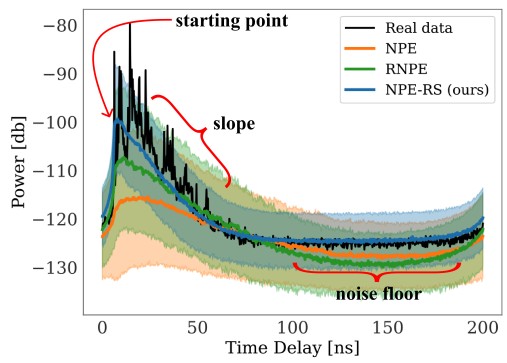

Figure 6: Predictive performance of NPE, RNPE and our NPE-RS method on the Turin model under misspecification. **NPE-RS leads to the best fit.** The shaded regions denote $\pm 1$ std. deviation of the posterior predictive distribution of each method.

We fit the model to the real data using NPE, RNPE and our NPE-RS methods. As there is no notion of ground truth in this case, we plot the resulting posterior predictive distributions in Figure 6. Note that the multiple peaks present in the data are not replicated by the Turin model for any method, which is due to the model being misspecified for this dataset. Despite that, our NPE-RS method appears to fit the data well, while NPE performs the worst. Moreover, our method is better than RNPE at matching the starting point of the data, the slope, and the noise floor, with RNPE underestimating all three aspects on average. The MMD between the observed data and the posterior predictive distribution of NPE, RNPE and NPE-RS is 0.11, 0.09, and 0.03, respectively. Hence, NPE-RS provides a reasonable fit of the model even under misspecification.

## 6 Conclusion

We proposed a simple and elegant solution for tackling misspecification of simulator-based models. As our method relies on learning robust statistics and not on the subsequent inference procedure, it is applicable to any SBI method that utilizes summary statistics. Apart from achieving robust inference under misspecified scenarios, the method performs reasonably well even when the model is well-specified. The proposed method only has one hyperparameter that encodes the trade-off between efficiency and robustness, which can be selected like other neural network hyperparameters, for instance, via a validation set.

---

[*]In wireless communications, the power of the signal is measured in decibels (dB). A power $P$ in linear scale corresponds to $10 \log_{10}(P)$ dB.

**Limitations and future work.** A limitation of our method is the increased computational complexity due to the cost of estimating the MMD, which can be alleviated using the sample-efficient MMD estimator from [10] or quasi-Monte Carlo points [56]. Moreover, as our method utilizes the observed statistic during the training procedure, the corresponding NPE is not amortized anymore — a limitation we share with RNPE. Thus, working on robust NPE methods which are still amortized (to some extent) is an interesting direction for future research. An obvious extension of the work could be to investigate if the ideas translate to likelihood-free model selection methods [68, 76] which can suffer from similar problems as SBI if all (or some) of the candidate models are misspecified.

## Acknowledgements

The authors would like to thank Dr. Carl Gustafson and Prof. Fredrik Tufvesson (Lund University) for providing the measurement data. We also thank Masha Naslidnyk, Dr. François-Xavier Briol, and Dr. Markus Heinonen for their useful comments and discussion. We acknowledge the computational resources provided by the Aalto Science-IT Project from Computer Science IT. This work was supported by the Academy of Finland Flagship programme: Finnish Center for Artificial Intelligence FCAI. SK was supported by the UKRI Turing AI World-Leading Researcher Fellowship, [EP/W002973/1].

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

# Supplementary Materials

The appendix is organized as follows:

- In Appendix B, we provide further implementation details and results for the numerical experiment in Section 4.

## A  Detecting misspecification of simulators

Considering that existing SBI methods can yield unreliable results under misspecification and that real-world simulators are probably not able to fully replicate observed data in most cases, detecting whether the simulator is misspecified becomes necessary for generating confidence in the results given by these methods. As misspecification can lead to observed statistics or features falling outside the distribution of training statistics, detecting for it essentially boils down to a class of out-of-distribution detection problems known as *novelty detection*, where the aim is to detect if the test sample $\mathbf{s}_{\mathrm{obs}}$ come from the training distribution induced by $\{s_i\}_{i=1}^m$. This two-label classification problem can potentially be solved by adapting any of the numerous novelty detection methods from the literature. We propose the following two simple novelty detection techniques for detecting misspecification:

**Distance-based approach.**   We assign a score to the observed statistic based on the value of the margin upper bound, as introduced in the main text. We use the MMD as the choice of distance $\mathcal{D}$, and estimate the MMD between the set of simulated statistics $\{s_i\}_{i=1}^m$ and the observed statistic $\mathbf{s}_{\mathrm{obs}}$. This MMD-based score can be used in a classification method to detect misspecification.

**Density-based approach.**   In this method, the training samples $\{s_i\}_{i=1}^m$ are used to fit a generative model $q$, and the log-likelihood of the observed statistics under $q$ are used as the classification score. We use a Gaussian mixture model (GMM) with $k$ components as $q$, having the distribution

$$q(s) = \sum_{i=1}^k \xi_i \varphi(s|\mu_i, \Sigma_i), \tag{11}$$

where $\xi_i$, $\mu_i$, and $\Sigma_i$ are the weight, the mean and the covariance matrix associated with the $i^{\mathrm{th}}$ component, and $\varphi$ denotes the Gaussian pdf. The score $\ln q(\mathbf{s}_{\mathrm{obs}})$ can then be used to classify it as either being from in or out of the training distribution.

**Experimental set-up.**   We test the performance of the proposed detection methods on the Ricker model and the OUP with the same contamination model as given in the main text. For each of these simulators, we first train the NPE method on $m = 1000$ training data points, and fit a GMM with $k = 2$ components to them. We then generate 1000 test datasets or points, half of them from the well-specified model and the other half from the misspecified model, and compute their score. The area under the receiver operating characteristic (AUROC) is used as the performance metric.

**Baseline.**   We construct a baseline for comparing performance of the proposed detection methods. The baseline is based on the insight that under model misspecification, the NPE posterior moves away from the true parameter value (even going outside the prior range). Therefore, we take the root mean squared error (RMSE), defined as $(1/N \sum_{i=1}^N (\theta_i - \theta_{\mathrm{true}})^2)^{\frac{1}{2}}$ where $\{\theta_i\}_{i=1}^N$ are posterior samples, as the classification score.

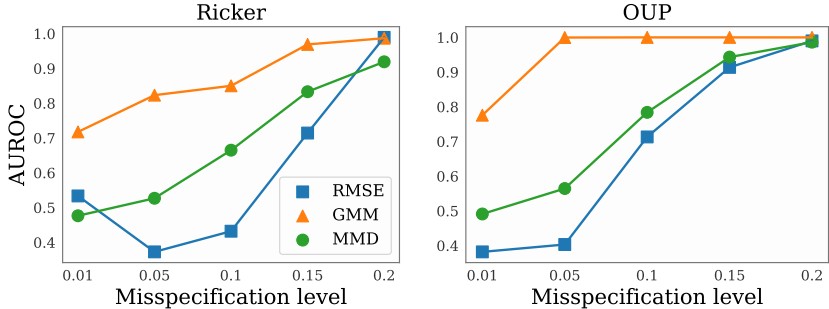

Figure 7: **Misspecification detection experiment**. AUROC of the proposed detection methods (GMM and MMD) versus misspecification level for the Ricker model and the OUP. The RMSE-based baseline is shown in blue.

**Results.** The AUROC of the classifiers for different levels of misspecification ($\epsilon$ in the main text) is shown in Fig. 7 for both the models. The proposed GMM-based detection method performs the best, followed by the MMD-based method. The RMSE-based baseline performs the worst at the classification task. We conclude that it is possible to detect model misspecification in the space of summary statistics using simple to use novelty detection methods.

# B  Additional details and results of the numerical experiments

## B.1  Implementation details

We implement our NPE-RS models based on publicly available implementations from `https://github.com/mackelab/sbi`. We use the NPE-C model [41] with Masked Autoregressive Flow (MAF) [60] as the backbone inference network, and adopt the default configuration with 50 hidden units and 5 transforms for MAF. The batch size is set to 50, and we maintain a fixed learning rate of $5 \times 10^{-4}$. The implementation for RNPE is sourced directly from the original repository at `https://github.com/danielward27/rnpe`.

Regarding the summary network in NPE tasks, for the Ricker model, we employ three 1D convolutional layers with 4 hidden channels, and we set the kernel size to 3. For the OUP model, we combine three 1D convolutional layers with one bidirectional LSTM layer. The convolutional layers have 8 hidden channels and a kernel size equal to 3, while the LSTM layer has 2 hidden dimensions. We pass the data separately through the convolutional layers and the LSTM layer and then concatenate the resulting representations to obtain our summary statistics. For the Turin model in Section 5, we utilize five 1D convolutional layers with hidden units set to [8, 16, 32, 64, 8], and the kernel size is set to 3. Across all three summary networks, we employ the `mean` operation as our aggregator to ensure permutation invariance among realizations.

In ABC tasks, we incorporate autoencoders as our summary network. For the Ricker model, the encoder consists of three 1D convolutional layers with 4 hidden channels, where the kernel size is set to 3. The decoder comprises of three 1D transposed convolutional layers with the same settings as the encoder's convolutional layers, allowing for data reconstruction. For the OUP model, we adopt a similar summary network as the one used for the Ricker model but with a smaller stride.

In NPE tasks, we use 1000 samples for the training data, along with 100 realizations of both observed and simulated data for each value of $\theta$. We also use 1000 samples for training the autoencoders. For ABC, we use 4000 samples from the prior and accept $n_\delta = 200$ samples giving a tolerance rate of 5%. We take $\rho$ to be Euclidean distance in the rejection ABC and normalize the statistics by the median absolute deviation before computing the distance to account for the difference in their magnitude.

## B.2  Additional posterior plots

We now present examples of the remaining posterior plots, apart from the one shown in the main text. The posterior plots for OUP using the NPE-based methods is shown in Figure 8. The observations

are similar to the Ricker model example in the main text: we see that our NPE-RS method yields similar posterior as NPE in the well-specified case, whereas RNPE posteriors are underconfident. When the model is misspecified, NPE posterior goes far from the true parameter value. The NPE-Rs posteriors, however, are still around $\theta_{\text{true}}$, demonstrating robustness to misspecification.

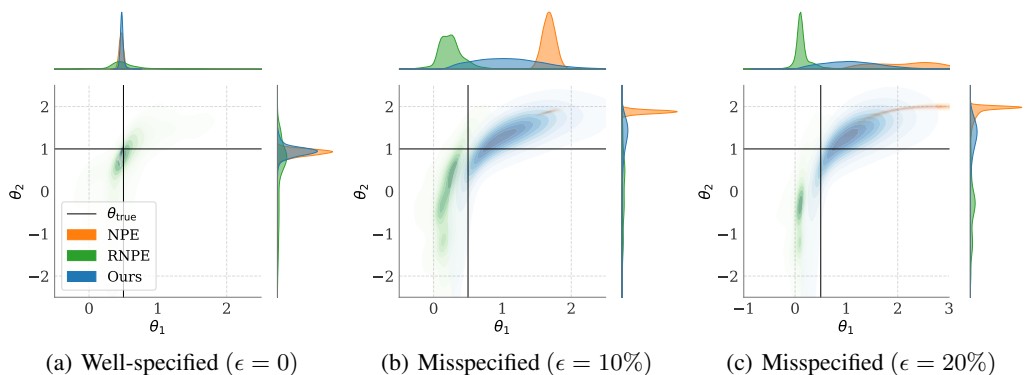

(a) Well-specified ($\epsilon = 0$)     (b) Misspecified ($\epsilon = 10\%$)     (c) Misspecified ($\epsilon = 20\%$)

Figure 8: **Ornstein-Uhlenbeck process.** Posteriors obtained from our method (NPE-RS), RNPE, and NPE for different degrees of model misspecification.

Similar behavior is observed in the ABC case for both the Ricker model and OUP in Figure 9 and Figure 10, respectively. The ABC posteriors go outside the prior range under misspecification, while ABC with our robust statistics yields posteriors closer to $\theta_{\text{true}}$. In Table 1, we report the sample mean and standard deviations for the results shown in Figure 2 of the main text.

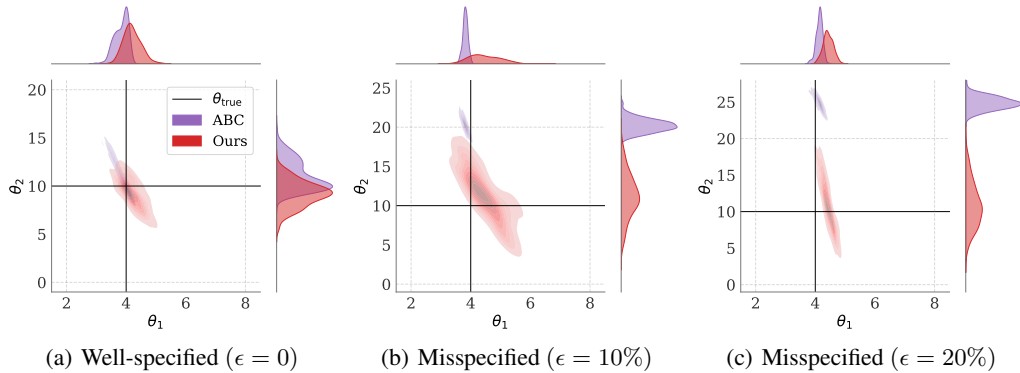

(a) Well-specified ($\epsilon = 0$)     (b) Misspecified ($\epsilon = 10\%$)     (c) Misspecified ($\epsilon = 20\%$)

Figure 9: **Ricker model.** Posteriors obtained from our method (ABC-RS) and ABC for different degrees of model misspecification.

### B.3   Results for $\mathcal{D}$ being the Euclidean distance

We present results for $\mathcal{D}$ being the Euclidean distance in the well-specified case of the Ricker model in Figure 11(a). As mentioned in Section 3 of the main text, this choice leads to very underconfident posteriors. This is because the Euclidean distance is not a robust distance: it becomes large even if a few points are far from the observed statistic. As a result, using this as the regularization term penalizes most choices of summarizer $\eta$, and we learn statistics that are very concentrated around the observed statistic (orange dot). Although a good choice for being robust, Euclidean distance leads to statistics that are not informative about the model parameters, yielding posterior that is similar to the uniform prior. Hence, we used the MMD as the distance in the margin upper bound, which provides a better trade-off between robustness and efficiency (in terms of learning about model parameters).

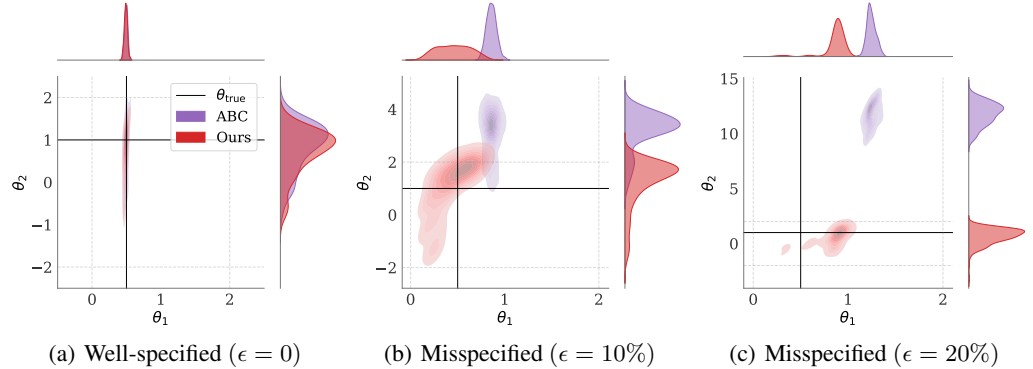

(a) Well-specified ($\epsilon = 0$)  (b) Misspecified ($\epsilon = 10\%$)  (c) Misspecified ($\epsilon = 20\%$)

Figure 10: **Ornstein-Uhlenbeck process.** Posteriors obtained from our method (ABC-RS) and ABC for different degrees of model misspecification.

Table 1: Performance of the SBI methods in terms of RMSE and MMD for both Ricker and OUP. We report the average ($\pm 1$ std. deviation) values across 100 runs for varying levels of misspecification.

|  |  | RMSE ($\downarrow$) | | | MMD ($\downarrow$) | | |
|---|---|---|---|---|---|---|---|
|  |  | $\epsilon = 0\%$ | $\epsilon = 10\%$ | $\epsilon = 20\%$ | $\epsilon = 0\%$ | $\epsilon = 10\%$ | $\epsilon = 20\%$ |
| Ricker | NPE | **2.16** (3.07) | 7.86 (1.57) | 11.2 (1.70) | **0.04** (0.07) | 0.74 (0.09) | 1.06 (0.17) |
|  | RNPE | 3.27 (0.35) | 5.51 (0.58) | 7.14 (1.15) | 0.06 (0.05) | 0.51 (0.19) | 0.79 (0.25) |
|  | NPE-RS (ours) | 2.18 (2.66) | **2.19** (1.01) | **4.66** (4.15) | 0.09 (0.14) | **0.21** (0.16) | **0.42** (0.37) |
|  | ABC | 1.46 (0.44) | 6.95 (0.25) | 9.79 (0.96) | 0.01 (0.01) | 0.85 (0.02) | 1.18 (0.04) |
|  | ABC-RS (ours) | **1.20** (0.51) | **3.16** (1.08) | **2.99** (1.28) | 0.01 (0.02) | **0.17** (0.15) | **0.18** (0.16) |
| OUP | NPE | 0.79 (0.62) | 1.26 (1.18) | 2.59 (2.75) | 0.01 (0.01) | 0.34 (0.15) | 0.63 (0.29) |
|  | RNPE | 0.78 (0.09) | 0.87 (0.10) | 0.98 (0.15) | 0.01 (0.01) | 0.22 (0.13) | 0.49 (0.26) |
|  | NPE-RS (ours) | **0.74** (0.70) | **0.62** (0.33) | **0.63** (0.36) | 0.02 (0.05) | **0.09** (0.09) | **0.21** (0.17) |
|  | ABC | 0.50 (0.07) | 1.20 (0.40) | 5.16 (2.39) | 0.05 (0.03) | 0.88 (0.21) | 0.92 (0.23) |
|  | ABC-RS (ours) | **0.44** (0.06) | **0.62** (0.23) | **0.88** (0.48) | **0.02** (0.02) | **0.26** (0.17) | **0.50** (0.38) |

## B.4 Computational cost analysis

We now present a quantitative analysis of the computational cost of training with and without our MMD regularization term. The results, presented in Table 2, are calculated on an Apple M1 Pro CPU. As expected, we observe a higher runtime for our method due to the computational cost of estimating the MMD from 200 samples of simulated data. The total runtime also depends on the number of batchsize $N_{\text{batch}}$, hence, as $N_{\text{batch}}$ increases, the proportion of runtime used for estimating MMD reduces. As a result, we see that for large $N_{\text{batch}}$, the increase in the computational cost of our method with robust statistics is not significant.

## B.5 Adversarial training on Gaussian linear model

To verify the robustness of our method on higher dimensional parameter space, we run an experiment on the Gaussian linear model, where the data $\mathbf{x} \in \mathbb{R}^{10}$ is sampled from $\mathcal{N}(\theta, 0.1 \cdot \boldsymbol{I}_{10})$. A uniform prior $\mathcal{U}([-1, 1])^{10}$ is placed on the parameters $\theta \in \mathbb{R}^{10}$. We take $\theta_{\text{true}} = [0.5, ..., 0.5]^\top$, $\theta_c = [2, ..., 2]^\top$. To introduce contamination to the observed data, we employ the same approach as outlined in the main text of our paper. However, there is a slight divergence in our experimental setup. In this example, we employ adversarial training, meaning that the model is trained on observed data with a high degree of misspecification ($\epsilon = 20\%$), while we perform inference on data with a lower misspecification degree ($\epsilon = 10\%$). For the summary network, we utilize the DeepSet [86] architecture. The encoder comprises two linear layers, each with a width of 20 hidden units, paired with the ReLU activation function. The decoder is constructed with a single linear layer of 20 hidden units.

The results are shown in Table 3. NPE-RS outperforms NPE in terms of MMD between the posterior predictive distribution and the observed data, which highlights the effectiveness of our approach in

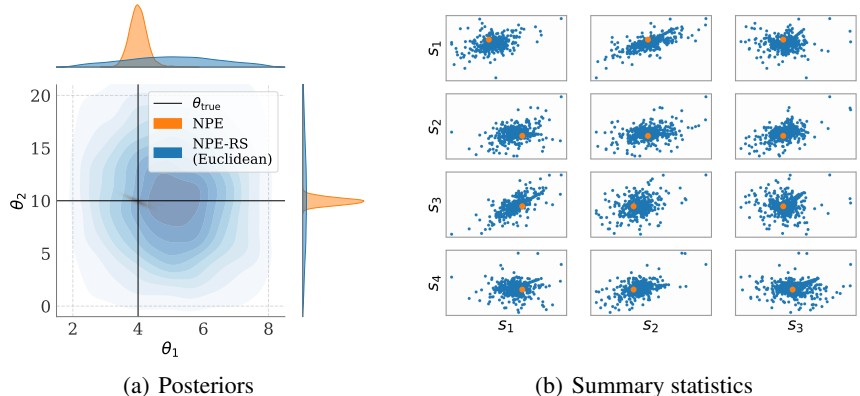

| (a) Posteriors | (b) Summary statistics |

Figure 11: **Ricker model.** Posteriors and summary statistics for $\mathcal{D}$ being the Euclidean distance.

Table 2: Comparison of computational costs across different models on Ricker model. We report the mean value (standard deviation) derived from 20 updates. We use different batch size $N_{\text{batch}}$ and generate 100 realizations for each $\theta$.

|  | Runtime (seconds) | | |
|---|---|---|---|
|  | $N_{\text{batch}} = 50$ | $N_{\text{batch}} = 100$ | $N_{\text{batch}} = 200$ |
| NPE | 0.22 (0.03) | 0.46 (0.04) | 0.87 (0.03) |
| NPE-RS (ours) | 1.26 (0.05) | 1.53 (0.14) | 1.92 (0.10) |
| ABC | 0.68 (0.04) | 1.41 (0.04) | 3.29 (0.27) |
| ABC-RS (ours) | 1.79 (0.04) | 2.71 (0.25) | 4.25 (0.46) |

high-dimensional parameter spaces, even though the observed data was not used in the training of NPE-RS. This points towards the possibility that by employing adversarial training, we might achieve robustness against lower levels of misspecification whilst still being amortized.

Table 3: Performance comparison between NPE and NPE-RS for Gaussian linear model. We use MMD between the posterior predictive distribution and the observed data as our metric. We report the average (±1 std. deviation) values across 100 runs.

|  | NPE | NPE-RS | | |
|---|---|---|---|---|
| $\lambda$ | - | 20 | 50 | 100 |
| MMD | 0.26 (0.02) | 0.19 (0.04) | **0.18** (0.06) | 0.21 (0.08) |

## B.6    Experiment with neural likelihood estimators

In this section, we explore the performance of our method when paired with Neural Likelihood Estimators (NLE). NLE are a class of methods that leverage neural density estimators to directly estimate likelihood functions, bridging the gap between simulators and statistical models.

For this experiment, we adopt NLE-A as proposed by [61]. The original implementation can be found at `https://github.com/mackelab/sbi`. Similar to our approach with ABC, we adapt our method to NLE by pre-emptively training an autoencoder with our regularization term to learn the summary statistics. We refer to our adapted method as NLE-RS. The configurations for our summary network and simulator are consistent with those described in Appendix B.1.

Figure 12 presents the posterior plots for the Ricker model using the NLE-based methods. Consistent with our observations in the previous experiments, NLE-RS still demonstrates robustness to misspecification, while the NLE posterior tends to deviate away from the true parameters.

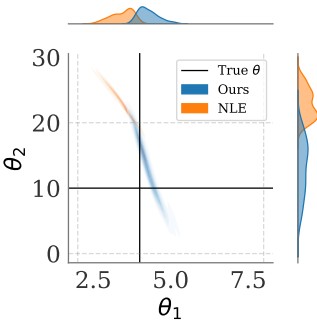

Figure 12: **Ricker model.** Posteriors obtained from our method (NLE-RS) and NLE. We set $\epsilon = 10\%$ for this experiment.

## C  Details of the radio propagation experiment

In this section, we describe the data and the Turin model used in Section 5 of the main text.

**Data and model description.**   Let $B$ be the frequency bandwidth used to measure radio channel data at $K$ equidistant points, leading to a frequency separation of $\Delta f = B/(K-1)$. The measured transfer function at $k$th point, $Y_k$, is modelled as

$$Y_k = H_k + W_k, \quad k = 0, 1, \ldots, K-1,$$

where $H_k$ is the transfer function at the $k$th frequency, and $W_k$ is additive zero-mean complex circular symmetric Gaussian noise with variance $\sigma_W^2$. Taking the inverse Fourier transform, the time-domain signal $y(t)$ can be obtained as

$$y(t) = \frac{1}{K} \sum_{k=0}^{K-1} Y_i \exp(j2\pi k \Delta f t).$$

The Turin model defines the transfer function as $H_k = \sum_l \alpha_l \exp(-j2\pi\Delta f k \tau_l)$, where $\tau_l$ is the time-delay and $\alpha_l$ is the complex gain of the $l^{\text{th}}$ component. The arrival time of the delays is modeled as one-dimensional homogeneous Poisson point processes, i.e., $\tau_l \sim \text{PPP}(\mathbb{R}_+, \nu)$, with $\nu > 0$. The gains conditioned on the delays are modeled as iid zero-mean complex Gaussian random variables with conditional variance $\mathbb{E}[|\alpha_l|^2|\tau_l] = G_0 \exp(-\tau_l/T)/\nu$. The parameters of the model are $\theta = [G_0, T, \nu, \sigma_W^2]^\top$. The prior ranges used for the parameters are given in Table 4.

Table 4: Prior distributions for the parameters of the Turin model.

|  | $G_0$ | $T$ | $\nu$ | $\sigma_W^2$ |
|---|---|---|---|---|
| Prior | $\mathcal{U}(10^{-9}, 10^{-8})$ | $\mathcal{U}(10^{-9}, 10^{-8})$ | $\mathcal{U}(10^7, 5 \times 10^9)$ | $\mathcal{U}(10^{-10}, 10^{-9})$ |

The radio channel data from [44] is collected in a small conference room of dimensions $3 \times 4 \times 3$ m$^3$, using a vector network analyzer. The measurement was performed with a bandwidth of $B = 4$ GHz, and $K = 801$. Denote each complex-valued time-series by $\tilde{\mathbf{y}} \in \mathbb{R}^K$, and the whole dataset by $\tilde{\mathbf{y}}_{1:n}$, where $n = 100$ realizations. We take the input to the summary network to be $\mathbf{y}_{1:n} = 10\log_{10}(|\tilde{\mathbf{y}}_{1:n}|^2)$.

**Scatter-plot of learned statistics.**   In Figure 13 and Figure 14, we show the scatter-plots of the learned statistics using the NPE and our NPE-RS method, respectively. We observe that the observed statistics (shown in orange) is often outside the set of simulated statistics (shown in blue) for the NPE method. Hence, the inference network is forced to generalize outside its training distribution, which leads to poor fit of the model, as shown in Section 5 of the main text. On the other hand, the observed statistic is always inside the set of simulated statistics (or the training distribution) for our method in Figure 14, which leads to robustness against model misspecification.

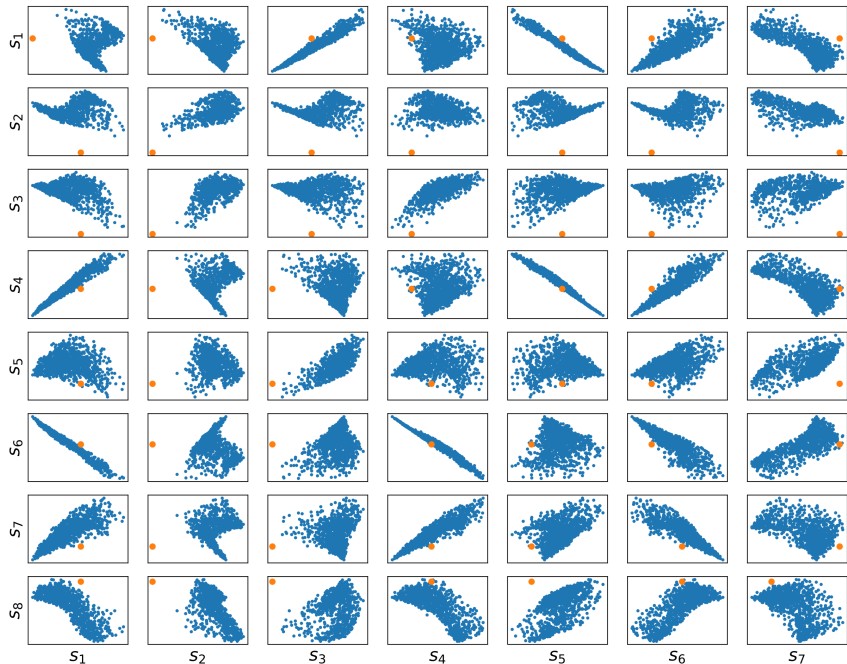

Figure 13: Pairwise scatter-plots of summary statistics learned using NPE method for the Turin model. Each blue dot corresponds to simulated statistic obtained from a parameter value sampled from the prior. The orange dot represents the observed statistic.

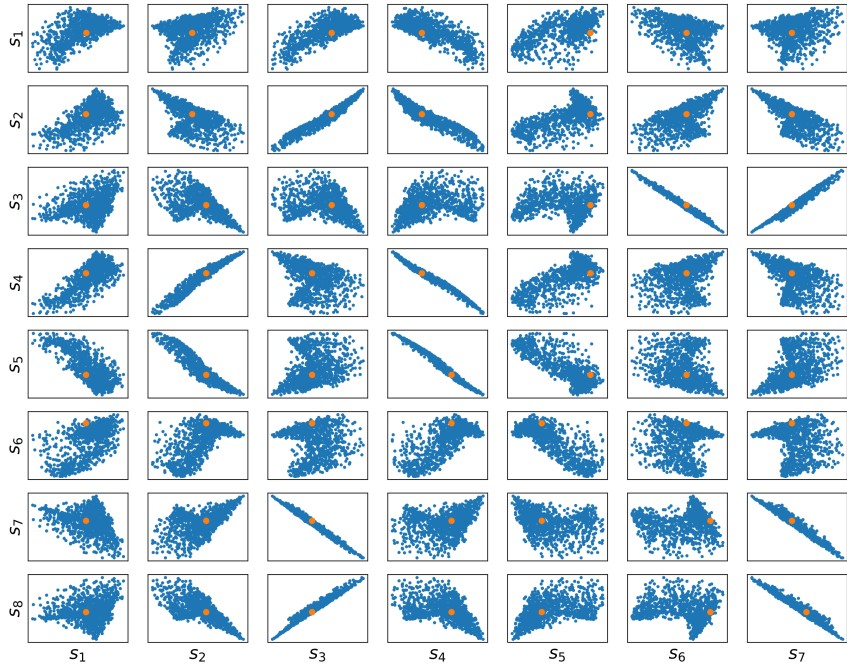

Figure 14: Pairwise scatter-plots of summary statistics learned using our NPE-RS method for the Turin model. Each blue dot corresponds to simulated statistic obtained from a parameter value sampled from the prior. The orange dot represents the observed statistic.

