# OpenReview forum: "Learning Robust Statistics for Simulation-based Inference under Model Misspecification"
_NeurIPS.cc/2023/Conference — NeurIPS 2023 poster_

### Official Review · Reviewer_1V4M · 2023-06-30

**Soundness:** 3 good
**Presentation:** 2 fair
**Contribution:** 2 fair
**Rating:** 5
**Confidence:** 3

**Summary:**

This paper proposes a general approach to handling model misspecification for SBI. The paper introduces a regularized loss function that penalises mismatches between the observed data and the learned model. The paper focuses on NPE and ABC as the likelihood-free approaches.

**Strengths:**

* The radio propagation example is interesting, and shows what the potential strength of using the MMD regularizer.
* The structure of the paper is clear.


**Weaknesses:**


* The novelty of the work is a concern in light of reference [72] equation (9), compared to equation (9) in this paper. Additionally, equation (10) appears similar to the InfoVAE (https://arxiv.org/abs/1706.02262), which also uses an auto-encoder with a MMD loss in the latent space.
* A minor weakness is that the paper is difficult to follow. While the structure made sense, the descriptions of the approach and the motivation for it were challenging to understand. For example, it seemed implicit that this approach did not perform amortised NPE until it was highlighted this was the case in the conclusion. This is not in itself a criticism of the approach, but it was very hard to follow what consisted of the training data, and observations. For example, line 255 mentions using 1000 samples for the training data and 100 realisations of both the observed and simulated data for each $\theta$. Under an amortised setting we would not have access to the observed $\theta$. It is not clear what this setup is from the description. Further questions in the next section highlight some of these confusions that likely stem from descriptions of the approach that could be improved.
* An additional minor weakness is only comparing to ABC and NPE. Is there any limitation on applying this approach to other estimators such as Neural Likelihood Estimators? Also for ABC, the $\rho$ was only described in the supplementary materials as the Euclidean distance. After defining the MMD as the better regularizer, why would this not also be used/incorporated into the discrepancy?

Typo: line 180, the $i$ should be subscript?

* In Figure 5, it would be useful to ensure the three plots share the same y axis.


**Questions:**

* Could the authors provide more details about the step in Eq. (4) to go from infimum to upper bound? I found this challenging to understand.
* The motivation behind defining a Q and a P that are different is a bit confusing. Why were these definitions highlighted? It seems that Q is equal to P when using the $\theta_{true}$, but is it a requirement to know $\theta_{true}$ prior to the experiments in order to simulate from Q?
* The last sentence of section 5 highlights that the MMD is superior for the NPE-RS method. Is this surprising given it is built into the loss function? Are there other statistics that could be used here that are specific to the application domain?


**Limitations:**

This is well captured by the paper. This is appreciated!

---

> ### Author Rebuttal · Authors · 2023-08-09
>
> * **"The novelty of the work is a concern in light of reference [72]":** See the general response to all the reviewers.
>
> * **"equation (10) appears similar to the InfoVAE (https://arxiv.org/abs/1706.02262), which also uses an auto-encoder with a MMD loss in the latent space.":** InfoVAE tackles the issue of learning meaningful latent features in variational autoencoders by modifying the ELBO objective, whereas we tackle the problem of learning robust statistics for simulation-based inference. Moreover, the MMD in InfoVAE is computed between variational distribution and prior distribution, while we compute the MMD between the simulated and observed statistics.
>
> * **"Is there any limitation on applying this approach to other estimators such as Neural Likelihood Estimators?":** No, our method can be applied to Neural Likelihood Estimators (NLEs) in the same way as we applied it to ABC: by learning a summary function up front using neural networks. See Figure 2 in the attached pdf for NLE results under misspecification.
>
> * **"After defining the MMD as the better regularizer, why would this not also be used/incorporated into the (ABC) discrepancy?":** MMD is used as the discrepancy in ABC to circumvent the need to summarize data into summary statistics, as mentioned in ref [61]. However, this requires the need to select an appropriate kernel function for the data, which is not always feasible. In most practical cases, summary statistics are used in ABC along with the Euclidean distance, which is what we used in our experiments. Euclidean distance is preferred here as the data is summarized into one statistic vector for each parameter value, while MMD is useful when computing distance between datasets.
>
> * **"line 255 mentions using 1000 samples for the training data and 100 realisations of both the observed and simulated data for each $\theta$. Under an amortised setting we would not have access to the observed $\theta$.":** We are not sure we understand. Did you mean "in the real-world setting we would not have access to the true $\theta$"? In that case, yes, that is the situation in the real-data experiment of Section 5. In section 4, we simulate observed data using the true $\theta$ to check if we are robust to misspecification in the parameter space.
>
> * **"Could the authors provide more details about the step in Eq. (4) to go from infimum to upper bound?":** Here we utilize the fact that the average value of a random variable (its mean) will be greater than or equal to its minimum value (infimum).
>
> * **"The motivation behind defining a Q and a P that are different is a bit confusing. Why were these definitions highlighted?":** We introduce $P_{\theta}$ and $Q$  to define model misspecification in SBI, see Section 2 of ref [24] for similar notation.
> Here $Q$ is the unknown true data-generating process, and $P_\theta$ is the model we are trying to fit. We use $\theta_{\mathrm{true}}$  in experiments to simulate the observed data. In practice, $\theta_{\mathrm{true}}$ is not known and we would only have samples from $Q$, which is the case in Section 5.
>
> * **"The last sentence of section 5 highlights that the MMD is superior for the NPE-RS method. Is this surprising given it is built into the loss function? Are there other statistics that could be used here that are specific to the application domain?":** The MMD in Section 5 is used to test the performance of the different methods based on their predictive distribution, and uses a different kernel than the MMD in the loss function that summarizes the data, thus they are unrelated. Nevertheless, by using the KL divergence estimator instead of MMD between the observed data and predictions, we get the values 4.59, 5.81 and 2.30 for NPE, RNPE and NPE-RS, respectively, showing again that our NPE-RS method fits the data the best.
>
> * **"In Figure 5, it would be useful to ensure the three plots share the same y axis."** Agreed. We will edit the figure accordingly.
> * **"Typo: line 180, the $i$ should be subscript?":** Yes, thanks for the careful reading.

---

> > ### Comment · Reviewer_1V4M · 2023-08-14
> > **Response**
> >
> > Thanks for your detailed response.
> >
> > > line 255
> >
> > I think my confusion comes from the difference between the $m$ training samples and $n$ realisations and how they both come into the algorithm.
> >
> > I am willing to raise my score, thanks to your response and additional experiments.

---

> > > ### Author Response · Authors · 2023-08-15
> > >
> > > Thank you for your response and for increasing your score. We really appreciate it.
> > >
> > > Regarding line 255: We have $n$ iid realisations for each dataset $\mathbf{x}\_{1:n} = \{\mathbf{x}^{(1)}\, \dots, \mathbf{x}^{(n)}\} \sim \mathbb{P}\_\theta$ simulated from the model for a given $\theta$. Thus, $(\theta, \mathbf{x}\_{1:n})$ form a pair. In order to train the NPE network (or for ABC), we generate $m$ such pairs by first sampling $\theta_1, \dots, \theta_m \sim p(\theta)$ from the prior, and then generating $\mathbf{x}\_{1:n, i} \sim \mathbb{P}\_{\theta_i}$, $1 \leq i \leq m$. This results in the training data $\{(\theta\_i, \mathbf{x}\_{1:n, i})\}_{i=1}^m$ of size $m$. So to sum up, there are $m$ simulated datasets in the training data, where each dataset has $n$ iid samples.

---

> > > > ### Comment · Reviewer_1V4M · 2023-08-15
> > > > **Thanks**
> > > >
> > > > That clears it up - thanks!

---

### Official Review · Reviewer_KhFo · 2023-07-02

**Soundness:** 2 fair
**Presentation:** 4 excellent
**Contribution:** 3 good
**Rating:** 5
**Confidence:** 3

**Summary:**

The authors propose a method to make neural posterior estimation robust to misspecification. The method relies on adding a regularizer to NPE loss function. The regularizer is implemented as the MMD between the embedding of the approximate (learned) posterior and the embedded observation. They apply their method to two low-dimensional benchmark tasks and a real-world example from radio propagation.

**Strengths:**

**Originality**:
The paper tackles an important issue: how to make neural simulation-based inference robust to misspecification. The method is novel and intuitive.

**Quality**:
The method is applied to benchmark tasks and to a real-world problem (with iid data samples and high-dimensional data).

**Clarity**:
The paper is well-written and easy-to-follow. Figures are intuitive and well-designed.

**Weaknesses:**

My main concerns with the paper are that it (1) uses only two very low-dimensional benchmark tasks and (2) overstates it contributions.

**Originality**:
- the method is a very straightforward extension to Schmitt et al 2021 (which is only cited with a passing reference). The authors should clearly discuss how the works are related.

**Quality**:
- The authors repeatedly emphasize that their method performs on par with NPE for well-specified models. I do not think that their results support this claim though. NPE-RS performs significantly worse than NPE on well-specified data on one of the (only) two benchmark tasks. In order to make this claim, the authors should use significantly more benchmark tasks, ideally with varying parameter and data-dimensionality, and demonstrate that the behaviour shown in figure 2, left is a rare exception. In addition, the authors would have to show that NPE-RS converges (at least very closely) to the true posterior with many simulations (see below).

- Convergence: The method proposed by the authors does no longer converge to the true posterior distribution for well-specified models, which further emphasizes that the method does not replace detection of misspecification. I would appreciate an investigation of how regularization strenght trades-off performance on well-specified data vs robustness to misspecification.

- how is the hyperparameter chosen? The authors claim that the regularizer can be selected with a validation set (L353). It is unclear to me how this would work. What would be the loss function used to assess performance on the validation set? In L231 the authors also say that this might be done based on the posterior predictive distribution, but I think this is tricky because it (1) requires (potentially many) more simulations and (2) can easily lead to pathological cases where the posterior is off but the predictive distribution is good. Please elaborate on how the hyperparameter should be set in practice.


**Questions:**

The authors claim that RNPE is not amortized. However, as far as I understand, RNPE does not require retraining for new data (yes, it requires MCMC, but this can be very fast, especially for low-d parameter space). Please clarify if I misunderstood this or clarify in your paper.

**Limitations:**

State more explicitly that the method does **not** converge to the true posterior.

---

> ### Author Rebuttal · Authors · 2023-08-09
>
> * **"The method is a very straightforward extension to Schmitt et al 2021.":** See the general response to all the reviewers.
>
> * **"... the paper uses only two very low-dimensional benchmark tasks ":** We respectfully disagree. While the tasks may be low dimensional in the number of parameters, they are very high dimensional in terms of data (100, 50, and 801 for Ricker, OUP and Turin model, respectively). Since misspecification occurs in the data space, we argue that the dimensionality of the data is more relevant to the problem under study. We would also like to point out that we added the arguably more important real-world case of misspecification (over additional synthetic benchmark examples).
>
> * **"NPE-RS performs significantly worse than NPE on well-specified data on one of the (only) two benchmark tasks.":** We disagree that NPE-RS performs *significantly* worse, and present additional results on different performance metrics in Figure 1 of the attached pdf. We see that the NPE-RS posterior is close to the NPE posterior in terms of MMD, and has comparable, if not better, empirical coverage.
>
> * **"Convergence: The method proposed by the authors does no longer converge to the true posterior distribution for well-specified models, which further emphasizes that the method does not replace detection of misspecification.":** Correct, our method does not converge to the true posterior. That is the price to pay to be robust to misspecification. What we claim is that our method achieves posterior consistency by leveraging Theorem 1 from Frazier et al. (2018), {Asymptotic Properties of ABC}, Biometrika, that says that as long as the statistics are informative about $\theta$ (which is the case for our method), the resulting ABC posterior concentrates on the true theta in the well-specified case. Unfortunately, such a result does not exist for NPE; however, as shown in Figure 1 of the attached pdf, the NPE-RS posterior is close to the NPE posterior in the well-specified case. Nevertheless, we agree that our results do not explicitly show that detecting misspecification is unnecessary, and we will remove that claim.
>
> * **"The authors would have to show that NPE-RS converges (at least very closely) to the true posterior with many simulations (see below).":** We do not claim that NPE-RS converges to the true posterior, but rather to the NPE posterior as $\lambda$ goes to 0, as shown in Figure 5(right).
>
> * **"I would appreciate an investigation of how regularization strength trades-off performance on well-specified data vs robustness to misspecification.":** This is exactly what we investigated in Figure 5 by varying the value of $\lambda$ in different settings.
>
> * **"How is the hyperparameter chosen? In L231 the authors also say that this might be done based on the posterior predictive distribution, but I think this is tricky because it (1) requires (potentially many) more simulations and (2) can easily lead to pathological cases where the posterior is off but the predictive distribution is good. Please elaborate on how the hyperparameter should be set in practice.":**
>     Our method only has one hyperparameter $\lambda$, which we propose to set either by using posterior predictive checks or via inference results on a held-out validation dataset (which can be a subset of the observed data), as it depends on the degree of misspecification in practice. As shown in Figure 5, we found the inference results to not be very sensitive to changes in $\lambda$ in various settings. We further argue that setting $\lambda$ is similar to setting other hyperparameters such as the choice of architecture, number of layers, choice of activation, learning rate etc., which is now an accepted part of fitting any deep learning-based model.
>
>     We argue that under model misspecification, it becomes difficult to say if the posterior is off or not (as there is no notion of a true posterior), as standard inference techniques are not reliable. We therefore rely on checking if the predictive distribution is accurate.
>
> * **"The authors claim that the regularizer can be selected with a validation set (L353). It is unclear to me how this would work. What would be the loss function used to assess performance on the validation set?":** We test the performance of the method on the validation set with different $\lambda$ values. The performance metric can be any loss function, such as MMD (which is what we used) or KL divergence, between the model's predictive distribution and the validation dataset. The same metric can also be computed on the space of statistics instead of the data.
>
> * **"The authors claim that RNPE is not amortized. However, as far as I understand, RNPE does not require retraining for new data (yes, it requires MCMC, but this can be very fast, especially for low-d parameter space). Please clarify if I misunderstood this or clarify in your paper.":** Agreed. Here, we refer to the amortization of the entire inference procedure, and not just the surrogate neural network, as noted in Box 1 of Lueckmann et al., Benchmarking Simulation-based Inference, AISTATS, 2021. Due to the additional MCMC step, the inference procedure is not amortized, which is also the case with neural likelihood and ratio estimators (NLE and NRE). We will clarify the text to reflect this distinction.

---

> > ### Comment · Reviewer_KhFo · 2023-08-13
> > **Response to rebuttal**
> >
> > Thanks for the detailed response and the additional simulations. They cleared some of my concerns, but some points are still unclear/unresolved to me:
> >
> > **Low-dimensionality:** I indeed had meant low-d theta. I agree that high-D x are relevant and important, but I still think that the paper would be much stronger if the authors demonstrated that the method scales to parameter spaces with dimensionality >> 2
> >
> > **NPE-RS performs significantly worse than NPE**: I still think that the paper downplays this limitation. The empirical difference between NPE and NPE-RS is very clearly noticable, also in the new results (in addition to NPE-RS not converging to the true posterior). In my opinion, claims like `It provides accurate inference results even when the model is well-specified` (L 67) are not warranted and have to be removed or significantly trimmed down.
> >
> > **trade-off between performance on well-specified data vs robustness to misspecification**: Which of the three metrics shown in figure 5 should correspond exactly to "robustness to misspecification" and why is this warranted? Wouldn't negative log-likelihood on misspecified x be a better measure?
> >
> > **Hyperparameter choice**: The authors say that it can be set with a `held-out validation dataset (which can be a subset of the observed data)`. Does this mean that using cross-validation requires several datasets of observed data (which are potentially misspecified and can not easily be synthetically generated) in order to choose $\lambda$? What if only one dataset is available at training time?

---

> > > ### Author Response · Authors · 2023-08-16
> > >
> > > Thank you for your response. We really appreciate it.
> > >
> > > * **"...I still think that the paper would be much stronger if the authors demonstrated that the method scales to parameter spaces with dimensionality $>>$ 2":** Agreed. Having an example with more than 4 parameters (as is the case with the Turin model in Section 5) can further strengthen our paper. To that end, we ran our method on the 10-dimensional Gaussian linear example with fixed covariance matrix $\Sigma$ (parameter of interest is the mean vector) used in the RNPE paper and also available in the SBI benchmark library. To make the model misspecified, we used the same contamination model for the observed data used in the paper, i.e., $\mathbb{Q} = (1-\epsilon)\mathcal{N}(\theta\_{\mathrm{true}}, \Sigma) + \epsilon \mathcal{N}(\theta\_c, \Sigma)$ where $\epsilon = 10$%, $\theta\_{\mathrm{true}} = [0.5, \dots, 0.5]^\top$, $\theta\_c = [2,\dots,2]^\top$, $p(\theta) = \mathcal{U}([-1,1])^{10}$. The average MMD between the posterior predictive distribution and the observed data over 100 runs is shown in the following table (std. deviation is reported in the parenthesis). We will of course include these results in the paper.
> > >
> > > |     |  NPE |     NPE-RS     |    NPE-RS    |     NPE-RS    |
> > > |:---:|:----:|:--------------:|:------------:|:-------------:|
> > > |     |   -  | $\lambda = 20$ | $\lambda=50$ | $\lambda=100$ |
> > > | **MMD** | 0.26 (0.02) |      0.19 (0.04)      |     **0.18** (0.06)     |      0.21 (0.08)     |
> > >
> > > * **"In my opinion, claims like 'It provides accurate inference results even when the model is well-specified' (L 67) are not warranted and have to be removed or significantly trimmed down":** Apologies if we weren't clear before. We agree that our results do not support the claim that "It provides accurate inference results even when the model is well-specified, thus circumventing any need to detect model misspecification", and we will remove it.
> > >
> > > * **"Wouldn't negative log-likelihood on misspecified x be a better measure?":** The log-likelihood is not applicable for simulator-based models due to the intractability of the likelihood function. We did consider fitting a multivariate Gaussian distribution to the predictive distributions from Ricker, OUP, and Turin model, and evaluating the log-likelihood of the data under the respective Gaussians. However, that introduces a new level of misspecification, as none of these models produce data that is jointly Gaussian. Moreover, when the observed data is corrupted by a few outlier points (thus causing model misspecification), the sum of log-likelihoods gets dominated by the few outliers, even if most of the observed data is explained well by the model. This is the reason why people have proposed generalised Bayesian inference (GBI) frameworks, where the likelihood term is replaced by a robust loss such as the MMD (which is known to be robust to outliers) to account for misspecification (see Knoblauch et al. (2019) and ref [24] for more details). We therefore used the MMD between the posterior predictive distributions and the observed data (bottom row of Figure 2) as the measure for robustness to misspecification (apart from RMSE).
> > >
> > > Knoblauch, J., Jewson, J., \& Damoulas, T. (2019). Generalized variational inference: Three arguments for deriving new posteriors. arXiv preprint arXiv:1904.02063.
> > >
> > > * **"Which of the three metrics shown in figure 5 should correspond exactly to "robustness to misspecification" and why is this warranted?":** In Figure 5, we are testing the claim that our method indeed converges to the NPE posterior as $\lambda$ goes to zero, and converges to the prior as $\lambda$ goes to infinity. To do that, we varied $\lambda$ and computed the distance (in terms of MMD) between the NPE-RS posterior and the prior (Fig 5(left)), and between the NPE-RS posterior and the NPE posterior (middle and right).
> > >
> > > * **Does this mean that using cross-validation requires several datasets of observed data (which are potentially misspecified and can not easily be synthetically generated) in order to choose $\lambda$? What if only one dataset is available at training time?:** We do not assume that multiple observed datasets are available, even though that is the case in some fields like Astrophysics where available data is plenty. We meant that for one observed dataset $\mathbf{y}\_{1:n} = \{\mathbf{y}^{(1)}\, \dots, \mathbf{y}^{(n)}\}$ with $n$ iid samples available at training time, we can use a random subset of the data, say $\mathbf{y}\_{1:m}$, where $m<n$, as the validation set to choose $\lambda$, and the rest of the $n-m$ points to run NPE-RS (which is what we did for the radio propagation experiment). In the limiting case that only one observed data point is available, there is not going to be any significant posterior update anyway.

---

> > > > ### Comment · Reviewer_KhFo · 2023-08-20
> > > > **Thank you!**
> > > >
> > > > Thank you for your response!
> > > >
> > > > I am still slightly leaning towards acceptance and will keep my initial score.

---

> > > > > ### Author Response · Authors · 2023-08-20
> > > > >
> > > > > Thank you for your time and consideration. We're glad to hear you're leaning towards acceptance. Please do let us know if there are any concerns remaining.

---

### Official Review · Reviewer_EXCq · 2023-07-04

**Soundness:** 2 fair
**Presentation:** 3 good
**Contribution:** 2 fair
**Rating:** 6
**Confidence:** 3

**Summary:**

This paper describes a method to perform simulation based inference, focusing on neural posterior estimation and approximate Bayesian computation (ABC), under model miss specification when performing inference using summary statistics of the data by using an MMD loss between the simulated and observed data summaries as a means to regulate miss specification.  Experiments are performed to empirically examine the efficacy of the method, and the method is compared to standard neural posterior estimation and ABC, as well as a method called robust neural posterior estimation.









**Strengths:**

The paper is tackling an important issue, the method is interesting, and the description of the method and its motivation are made clear.


**Weaknesses:**

It is not clear how this work differs from ref [72], Schmitt et. al, "Detecting Model Misspecification in Amortized Bayesian Inference with Neural Networks". Much clearer description of the differences is needed to understand the novelty of this work

It is not clear how to determine the amount of regularization, i.e. how to set the hyperparameter lambda. This seems to be a key missing piece of information on how to practically use this method.

The paper seems to indicate that the summarizer acts on sets of iid samples. But is it not clear why must the summarizer act on the set of x's, rather than summarizing each observation, and using the fact that the summaries will remain iid. This seems to come back as a constraint later in line 179, but does not seem well motivated, or practically how people use summary statistics for iid data. This could also significantly impact the ability to train a conditional normalizing flow due to the massive reduction in information by summarizing over a set of examples rather that summarizing per example.

In the experiments, it is not clear what summary statistic is used for RNPE? Summary statistics are not described in the RNPE paper, but rather attempting to model the p(theta|x), so how was this choice made? Does it affect the results?

Moreover, in the experiments, why not compare to the method in ref [72]?

In terms of reference, there are also methods to learn robust summaries prior to inference, e.g. using pivots, such as in Louppe, et. al, "Learning to pivot with adversarial networks", and similar domain adaptation approaches, some specifically using MMD.

**Questions:**

How does this work differ from ref [72] ? Can you provide a much clearer description of the differences is needed to understand the novelty of this work. Can you compare to this work?

On line 257, is is stated that lambda is set using simulations using theta_true. What does it mean that you set lambda using a data set with known theta_true? isn't this something that needs to be estimated? Doesn't this greatly reduce the challenge of inference, and this information is not realistically available?

Why does the summarizer act on on sets of iid data, rather than on each iid data example? Acting on each example individually seems to be much closer to practical usage. Do you have experiments in this setting? Would this significantly impact the computation driven by MMD, it seems this may make the method impractical?

Please provide more details on how RPNE was used in the experiments, and how the summary statistics were determined.



**Limitations:**

The authors adequately discuss limitations

---

> ### Author Rebuttal · Authors · 2023-08-09
>
> * **"How does this work differ from ref [72] ? Can you provide a much clearer description of the differences is needed to understand the novelty of this work. Can you compare to this work?":** See the general response to all the reviewers.
>
> * **``It is not clear...how to set the hyperparameter lambda'':** We propose to set the hyperparameter $\lambda$ either by using posterior predictive checks or via inference results on a held-out validation dataset (which can be a subset of the observed data). As shown in Figure 5, we found the inference results to not be very sensitive to changes in $\lambda$ in various settings. We further argue that setting $\lambda$ is similar to setting other hyperparameters such as the choice of architecture, number of layers, choice of activation, learning rate etc., which is now an accepted part of fitting any deep learning-based model.
>
> * **"Why does the summarizer act on sets of iid data, rather than on each iid data example? Acting on each example individually seems to be much closer to practical usage. Do you have experiments in this setting? Would this significantly impact the computation driven by MMD, it seems this may make the method impractical?":** We do not specify explicitly whether the summarizer acts on sets of iid data or on each iid data example. In fact, the latter is a special case of the former. The summarizer can act on each data example as well, however, that means there are as many summaries as number of data points, thus invoking the curse of dimensionality when computing distances in ABC. Moreover, computing statistics on the whole dataset can be more informative about parameters that govern the distributional behaviour of the data. We refer to [68] that define the summarizer in the same general way as us. For the experiments shown in the paper, the summary network we used first summarizes each data example, and then aggregates them into a single statistics vector for each $\theta$, similar to [68].
>
> * **"Summary statistics are not described in the RNPE paper, but rather attempting to model the $p(\theta|\mathbf x)$, so how was this choice made?":** We respectfully disagree. The RNPE paper does use summary statistics as $\mathbf{x}$, see the sentence before Section 2.2 in the RNPE paper that says "Hereafter, we perform model criticism and inference using handcrafted summary statistics...", and also Section 4.2 where they mention the statistics used for SIR and CS tasks.
>
> * **"Please provide more details on how RPNE was used in the experiments, and how the summary statistics were determined.":** The summary statistics learned using the joint NPE framework were the ones used for RNPE. That is, we used the output of the trained summary network in NPE as statistics of RNPE, thus making sure that both the methods used the same statistics so that comparing them was fair.
>
> * **"In terms of reference, there are also methods to learn robust summaries prior to inference, e.g. using pivots, such as in Louppe, et. al, "Learning to pivot with adversarial networks", and similar domain adaptation approaches, some specifically using MMD.":** Good point. We have made a note of it in the related works.
>
> * **"What does it mean that you set lambda using a data set with known $\theta_{\mathrm{true}}$?":** We do not assume the method knows the true theta; $\theta_{\mathrm{true}}$ simply is notation for the $\theta$ used for simulating the observed and validation data. We use the validation set, generated from true $\theta$, to set the value of $\lambda$.

---

> > ### Comment · Reviewer_EXCq · 2023-08-15
> > **Response**
> >
> > Dear Author, thank you for your detailed responses. Some of my concerns have been alleviated, and some of my confusions cleared up.
> >
> > - In general I think  would be useful to readers to ensure that the text clearly describes the differences to ref [72], as you have discussed in your responses. I think your response was helpful
> >
> > - Indeed, with a summarizer on iid data points, you only gain in the reduction of dimensionality from the data to summary size.  It was not clear why this would necessarily be a problem in ABC if summary is low dimensional. It was also not clear to me how a summarizer over datasets could handle data sets containing millions of iid samples, as is common in mane scientific applications. Thus summarizing over datasets introduces a different challenge of dimensionality.  Nonetheless, I recognize that this is not the main goal of this work, but rather the focus is on learning robust statistics, and not about  exactly how one summarizes the data.
> >
> > - I suggest changing notation, as theta_true is highly misleading. I am also not quite sure I understand your response. I suggest the authors improve the text here.

---

> > > ### Author Response · Authors · 2023-08-16
> > >
> > > Thank you for your response and for increasing your score. We really appreciate it.
> > >
> > > * **"In general I think would be useful to readers to ensure that the text clearly describes the differences to ref [72], as you have discussed in your responses. I think your response was helpful":**  We are glad. And yes of course we will include that text in the paper.
> > >
> > > * **I suggest changing notation, as theta_true is highly misleading. I am also not quite sure I understand your response. I suggest the authors improve the text here.":** For simulation experiments, the observed data is generated from some ground-truth parameter values, which we refer to as $\theta_{\mathrm{true}}$. This is done in order to measure the performance of the inference methods in the parameter space (see e.g. Figures 2, 4 and 5 in ref [41] where the ground truth parameters are denoted by red lines). As our method requires a validation dataset to set the hyperparameter $\lambda$, we use the same ground truth parameters to generate it as well. We will edit the notation and the text in the paper to clarify this point.

---

### Official Review · Reviewer_DQRY · 2023-07-05

**Soundness:** 3 good
**Presentation:** 3 good
**Contribution:** 3 good
**Rating:** 4
**Confidence:** 1

**Summary:**

The paper addresses the "data selection problem", i.e. identify a low-dimensional statistic (e.g. mean and variance for Gaussian distribution) of a high-dimensional dataset for which the model can replicate even when misspecified. The paper propose using the auto-encoding framework to automatically extract statistics and a penalty on mismatched statistics, i.e. statistics that the model is unable to replicate, as a general approach to handle model misspecification. Empirical results show robust inference in misspecified scenarios whilst still being accurate in well-specified scenarios.

**Strengths:**

1. The paper introduces the "data selection problem". Sorry, I am not a subject matter expert in the field of "data selection" thus I cannot discuss on the originality and significance of this work. However, I learnt about the data selection problem from the paper.

**Weaknesses:**

1. The ideas of auto-encoding, reducing mismatch and tuning the regularizer are not new, but they are perhaps new in the application to "data selection". Please discuss further.
2. From the paper, since it is an auto-encoding framework on the model, the statistics are informative of the model. On the other hand, how to ensure the statistics are informative of the observations? Please discuss further.

**Questions:**

1. Since the extracted statistics are changing/ unknown beforehand, how will these statistics fit into downstream applications for downstream applications that were fixed beforehand? Perhaps the downstream tasks are retrained on the new robust statistics.
2. In the experiment implementation, should the model be the misspecified case while the observations be the true case? Or both can have some degree of misspecification while we try to extract the true parameters?

**Limitations:**

1. Authors mentioned the limitation of using the observed statistic during the training procedure and recommended working on it as future work.

---

> ### Author Rebuttal · Authors · 2023-08-08
>
> * **"The ideas of auto-encoding, reducing mismatch and tuning the regularizer are not new, but they are perhaps new in the application to "data selection". Please discuss further.":** Yes, our contribution is to show how to solve the important problem of simulation-based inference under model misspecification by framing it as a data selection problem. As models in real world are often misspecified, addressing this problem is necessary for the application of the inference methods in many fields.
>
> * **"From the paper, since it is an auto-encoding framework on the model, the statistics are informative of the model. On the other hand, how to ensure the statistics are informative of the observations?":** Statistics are functions of the observations, and should be informative about the model parameters to perform inference. The first term in the loss function makes sure the statistics are informative, while the regularizer term ensures they are robust.
>
> * **"Since the extracted statistics are changing/ unknown beforehand, how will these statistics fit into downstream applications for downstream applications that were fixed beforehand? Perhaps the downstream tasks are retrained on the new robust statistics.":** Yes, the new robust statistics can be used for the downstream tasks. It is a good point that if the downstream application is known beforehand, the statistics can be chosen according to that task. To do that, the first term in Equations 9 and 10 can be replaced by a term that minimises the downstream loss. Thank you for pointing this out; we will add a comment about this to the Discussion.
>
> * **"In the experiment implementation, should the model be the misspecified case while the observations be the true case?:**  A model is misspecified or well-specified with respect to a given set of observations, which in the simulated experiments we have chosen to either match or not match the modelling assumptions. So yes, the observations are always the ’true case’.

---

> > ### Comment · Reviewer_DQRY · 2023-08-15
> >
> > Thanks for the response. I have read all the other reviews and responses. My score remains.

---

> > > ### Author Response · Authors · 2023-08-16
> > >
> > > Thank you for your time. Do let us know if there are any concerns which we can address.

---

> > > > ### Comment · Reviewer_DQRY · 2023-08-19
> > > >
> > > > Reduced the confidence level of my rating to 1 since I am not in this area. Hope this helps.

---

> > > > > ### Author Response · Authors · 2023-08-19
> > > > >
> > > > > Thanks a lot. That's very kind of you.

---

### Author Rebuttal · Authors · 2023-08-09

We thank the reviewers for their careful consideration. The reviewers agreed on the following strengths of the paper:
* **Relevance:** Reviewers EXCq and KhF0 agree that the ''paper is tackling an important issue''
* **Good presentation:** EXCq: ''description of the method is clear''. KhF0: ''The paper is well-written and easy-to-follow. Figures are intuitive and well-designed''. 1V4M: ''structure of the paper is clear''.
* **Methodology:** KhF0: ''method is novel and intuitive''. EXCq: ''method is interesting''.
* **Real-world experiment:** KhF0: ''method is applied to real-world problem with high-dimensional data''. 1V4M: ''radio propagation example is interesting, and shows the potential strength of using the MMD regularizer''.

The reviewers highlight concern regarding the novelty of the work in light of reference [72], Schmitt et. al, ``Detecting Model Misspecification in Amortized Bayesian Inference with Neural Networks'', which we clarify here.
* **The purposes of the methods are different:** while we tackle the problem of robust inference under model misspecification, [72] addresses the problem of detecting model misspecification.
* **Role of the regularizer:** we include the regularizer to learn a summarizing function that ensures that the observed statistic is not an out-of-distribution sample in the summary space. On the other hand, [72] proposes to add an MMD regularizer term between the simulated statistics and samples from a standard Gaussian, which ensures that the learned statistics are jointly Gaussian.
* **How statistics are used:** we use the learned statistics to perform inference, while [72] conducts a goodness of fit test of Gretton et al, (2012) to detect if the model is misspecified.
* **The scope is different:** while their method is only applicable for NPE, our method can be used to perform robust inference in other SBI methods as well, such as ABC, NLE, NRE.

As [72] does not provide a method for robust inference, comparing our method with theirs is infeasible. Their method is complementary to ours, such that our method can be used once misspecification has been detected using [72]. This is the reason why in the paper we compared our method to RNPE, which addresses the same problem of robust inference as we do.

We also include additional results in the attached pdf document in response to some of the questions raised by reviewers KhFo and 1V4M.
We address all the individual comments of the reviewers in separate responses to each.

---

### Decision · Program_Chairs · 2023-09-21

**Decision:**

Accept (poster)

**Comment:**

This paper an important problem in simulation based inference: how to handle misspecification since inferences are based on the assumption that the model is correct. All reviewers agree that the proposed method addresses this issue through the use of carefully constructed summary statistics. However, it is crucial that the authors adjust their paper to address the points made during the discussion phase.